# TrajMamba: An Efficient and Semantic-rich Vehicle Trajectory Pre-training Model

**Yichen Liu**[1][*]**, Yan Lin**[2][*]**, Shengnan Guo**[1,3]**, Zeyu Zhou**[1]**, Youfang Lin**[1,4]**, Huaiyu Wan**[1,4][†]

[1]School of Computer Science and Technology, Beijing Jiaotong University, China
[2]Department of Computer Science, Aalborg University, Denmark
[3]Key Laboratory of Big Data & Artificial Intelligence in Transportation, Ministry of Education, China
[4]Beijing Key Laboratory of Traffic Data Mining and Embodied Intelligence, China
`{liuyichen, guoshn, zeyuzhou, yflin, hywan}@bjtu.edu.cn,`
`lyan@cs.aau.dk`

## Abstract

Vehicle GPS trajectories record how vehicles move over time, storing valuable travel semantics, including movement patterns and travel purposes. Learning travel semantics effectively and efficiently is crucial for real-world applications of trajectory data, which is hindered by two major challenges. First, travel purposes are tied to the functions of the roads and points-of-interest (POIs) involved in a trip. Such information is encoded in textual addresses and descriptions and introduces heavy computational burden to modeling. Second, real-world trajectories often contain redundant points, which harm both computational efficiency and trajectory embedding quality. To address these challenges, we propose **TrajMamba**, a novel approach for efficient and semantically rich vehicle trajectory learning. TrajMamba introduces a *Traj-Mamba Encoder* that captures movement patterns by jointly modeling both GPS and road perspectives of trajectories, enabling robust representations of continuous travel behaviors. It also incorporates a *Travel Purpose-aware Pre-training* procedure to integrate travel purposes into the learned embeddings without introducing extra overhead to embedding calculation. To reduce redundancy in trajectories, TrajMamba features a *Knowledge Distillation Pre-training* scheme to identify key trajectory points through a learnable mask generator and obtain effective compressed trajectory embeddings. Extensive experiments on two real-world datasets and three downstream tasks show that TrajMamba outperforms state-of-the-art baselines in both efficiency and accuracy.

## 1 Introduction

A vehicle GPS trajectory is a sequence of (location, time) pairs recording the movement of the vehicle during its trip from one location to another. Recent progress in intelligent traffic systems (ITS) has highlighted the value of these trajectories in revealing travel semantics, including movement patterns and travel purposes. These semantic insights support various spatio-temporal data mining tasks, including trajectory prediction [38, 40, 3, 5], travel time estimation [45, 24, 34], anomaly detection [26, 15], trajectory similarity measurement [43, 17], and trajectory clustering [44]. With the increasing availability of vehicle GPS trajectories, a powerful trajectory learning model that can extract travel semantics with high efficiency becomes more and more essential in building ITS.

For the vehicle trajectory $\langle \tau_1, \tau_2, \ldots, \tau_5 \rangle$ in Fig. 1, where each point represents a GPS coordinate at a specific time and lies on one of four road segments $e_1$, $e_2$, $e_3$, and $e_4$. Points $\tau_2$ and $\tau_3$ share the same

---

[*]Both authors contributed equally to this research.
[†]Corresponding author.

road segment $e_2$. Both GPS coordinates and road segments provide spatial information about the vehicle's position. Together, they reveal continuous movement patterns (e.g., turning, accelerating, going straight), characterizing the vehicle's travel from one location to another.

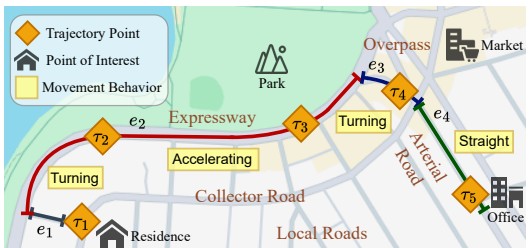

Figure 1: A trajectory of commuting to work.

Beyond movement patterns, travel purposes can be revealed by the functionalities of nearby road segments and POIs along a trajectory. For example, the trajectory in Fig. 1 starts in a residential area, follows an expressway, an overpass, and an arterial road, passes by a park and a market, and ends at an office building, indicating a commuting purpose.

Considering that many real-world applications of trajectories demand precise and real-time decisions, it is desirable to develop a trajectory learning method that effectively and efficiently incorporates the above information. However, this goal is hindered by two key challenges.

**First, extracting travel purposes can be computationally heavy.** As discussed, the functions of POIs and roads traversed by a trajectory indicate its travel purposes, but extracting these functions often requires incorporation of textual modality. Language models (LMs) [8, 1, 10] can capture these functions from POI and road descriptions, a capability explored in a few recent trajectory learning studies [47].However, integrating LMs into standard trajectory learning models significantly increases computational costs, since LMs are usually several magnitudes larger than standard trajectory learning models.

**Second, redundant points in trajectories affect both efficiency and effectiveness.** Vehicle GPS trajectories are usually sampled at a high frequency. This leads to real-world trajectories that often include redundant points, reducing encoding efficiency. For instance, points gathered during traffic stops or when vehicles maintain steady speeds may not provide useful information. Hence, it is beneficial to compress trajectories to improve both efficiency and effectiveness. Existing methods for trajectory compression mostly rely on rule-based or geometric approaches (e.g., Douglas-Peucker [9], Visvalingam-Whyatt [37]), which have high time complexities. A more efficient and learnable trajectory compression approach remains an open problem.

To overcome these issues, we propose *Trajectory Mamba* (**TrajMamba**), a new method for efficient and semantic-rich vehicle trajectory learning. TrajMamba introduces three components to achieve its design goal. First, a *Traj-Mamba encoder* jointly models GPS and road perspectives of trajectories. Second, a *travel purpose-aware pre-training procedure* that integrates travel purposes into the trajectory embeddings without introducing extra overhead to the embedding calculation. Third, a *knowledge distillation pre-training strategy* that locates key trajectory points through a learnable mask generator to reduce redundancy and obtains effective compressed trajectory embeddings for downstream tasks. We conduct extensive experiments on two real-world datasets and three downstream tasks, showing that TrajMamba outperforms state-of-the-art baselines in both accuracy and efficiency. Our robust outcomes in these three tasks exhibit average enhancements of 18.28%, 27.89%, and 17.68%, respectively.

## 2 Related Work

Vehicle trajectory learning methods extract valuable information from vehicle trajectories to perform various tasks. These methods can be broadly categorized into end-to-end trajectory learning methods and pre-trained trajectory embeddings.

**End-to-end Trajectory Learning Methods** are tailored for one specific task and are typically trained with task-specific labels. According to different task types, end-to-end methods can be roughly divided into: trajectory prediction methods [11, 19, 28, 40], trajectory classification methods [21, 4, 33], and trajectory similarity measurement methods [42, 46]. While end-to-end methods are straightforward to implement and offer certain advantages, they cannot be easily reused for other tasks. This necessitates designing, training, and storing separate models for each task, which can impact computational

resources and storage efficiency. Additionally, the effectiveness of end-to-end methods depends on the abundance of task-specific labels, which cannot always be guaranteed.

**Pre-trained Trajectory Embeddings** that can be utilized across various tasks have recently received increasing attention due to their ability to address the limitations of end-to-end methods. This approach involves learning trajectory encoders that map vehicle trajectories into embedding vectors, which can then be used with prediction modules. Earlier works [44, 20, 12] commonly leveraged RNNs to reconstruct discrete or continuous trajectory features based on auto-encoding frameworks [16], while methods like CTLE [23] and Toast [6] regard trajectory points as tokens in a sentence and process them using Transformers [36] and Masked Language Model (MLM) tasks [8]. TrajCL [2] and MMTEC [22] introduce contrastive learning [30] to train the models. Furthermore, recent methods combine multiple tasks to enhance their learning capability. START [18] and JGRM [27] leverage both MLM tasks and contrastive tasks, while LightPath [41] integrates a reconstruction task with a contrastive task.

Despite the promising progress made by existing efforts, as discussed in Section 1, there are still challenges in efficiently extracting travel purposes and removing redundant and noisy points in a learnable manner from vehicle trajectories due to their inherent complexity.

## 3 Preliminaries

**Road Network** A road network is modeled as a directed graph $\mathcal{G} = (\mathcal{V}, \mathcal{E})$. Here, $\mathcal{V}$ is a set of nodes, with each node $v_i \in \mathcal{V}$ representing an intersection between road segments or the end of a segment. $\mathcal{E}$ is a set of edges, with each edge $e_i \in \mathcal{E}$ representing a road segment linking two nodes. An edge is defined by its starting and ending nodes, and a textual description including the name and type of the road: $e_i = (v_j, v_k, \text{desc}_i^{\text{Road}})$.

**Vehicle Trajectory** A vehicle trajectory is a sequence of timestamped locations, defined as $\mathcal{T} = \langle \tau_1, \tau_2, \ldots, \tau_n \rangle$, where $n$ is the number of points. Each point $\tau_i = (g_i, e_i, t_i)$ consists of the GPS coordinates $g_i = (\text{lng}_i, \text{lat}_i)$, road segment $e_i$, and timestamp $t_i$, representing the vehicles location at a specific time. In this study, we use the term trajectory to refer to vehicle trajectory.

**Point of Interest** A point of interest (POI) is a location with specific cultural, environmental, or economic importance. We represent a POI as $p_i = (g_i, \text{desc}_i^{\text{POI}})$, where $g_i$ represents its coordinates, and $\text{desc}_i^{\text{POI}}$ is a textual description that includes its name, type, and address.

**Problem Statement** Vehicle trajectory learning aims to construct a learning model $f_\Theta$, where $\Theta$ is the set of learnable parameters. Given a vehicle trajectory $\mathcal{T}$, the model calculates its embedding vector as $e_\mathcal{T} = f_\Theta(\mathcal{T})$. This embedding vector $e_\mathcal{T}$ captures the travel semantics of $\mathcal{T}$ and can be used in subsequent applications by adding prediction modules.

## 4 Methodology

Fig. 2 provides the overall framework of TrajMamba, and its pipeline is implemented in the following three steps: 1) Given a trajectory $\mathcal{T}$, we introduce a Traj-Mamba Encoder to generate its embedding vector to effectively capture movement patterns. 2) To efficiently perceive travel purposes, we develop Travel Purpose-aware Pre-training to train the Traj-Mamba encoder by aligning the learned embedding with the road and POI views of $\mathcal{T}$, which encode the travel purpose through road and POI textual encoders. After this pre-training, we fix the weights of the encoder and regard it as the teacher model for the next step. 3) To effectively reduce redundancy in $\mathcal{T}$, we apply the Knowledge Distillation Pre-training. It employs a learnable mask generator to identify key trajectory points in $\mathcal{T}$ for compression, then aligns the compressed representation from a teacher-initialized Traj-Mamba encoder with the full-trajectory embedding from the teacher model.

### 4.1 Traj-Mamba Encoder

To effectively capture movement patterns, we design the Traj-Mamba Encoder consisting of $L$ stacked *Traj-Mamba blocks* inspired by the Mamba2 structure [7]. As shown in Fig. 3, each block employs two multi-input selective SSMs, namely GPS-SSM and Road-SSM, to capture long-term spatiotemporal

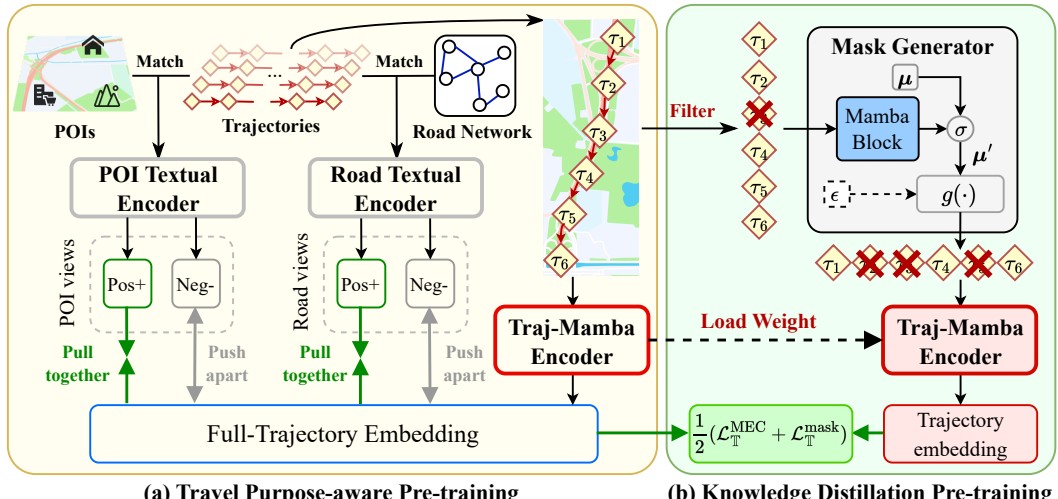

(a) Travel Purpose-aware Pre-training      (b) Knowledge Distillation Pre-training

Figure 2: The framework of TrajMamba.

correlations in input trajectories with linear time complexity. This enables the encoder to jointly model GPS and road perspectives and achieve the interaction between them to fuse information.

Given a trajectory $\mathcal{T} = \langle \tau_1, \tau_2, \ldots, \tau_n \rangle$, we first extract three features from each point for encoding its GPS perspective, including GPS coordinates, the time delta $\Delta t_i$ relative to $t_1$, and the timestamp in minutes. For the road perspective, we extract four features, including the road ID, day-in-week, hour-in-day, and minute-in-hour of each point. We then embed these features to obtain GPS and road latent vector sequences $\boldsymbol{Z}_{\mathcal{T}}^G, \boldsymbol{Z}_{\mathcal{T}}^R \in \mathbb{R}^{n \times \frac{E}{2}}$ as the inputs $\boldsymbol{Z}_0^G, \boldsymbol{Z}_0^R$ to the first Traj-Mamba block, where $E$ represents the dimension of trajectory embeddings (see Appendix H for details). To model continuous movement behavior, we also compute the speed, acceleration, and movement angle of each point,

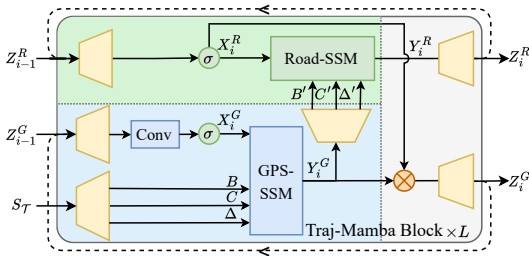

Figure 3: Structure of Traj-Mamba blocks.

which form a high-order movement feature sequence $\boldsymbol{S}_{\mathcal{T}} \in \mathbb{R}^{n \times 3}$. Next, to explain the Traj-Mamba block, we use the $i$-th block as a representative example.

In the **GPS-SSM branch**, we first transform the input $\boldsymbol{Z}_{i-1}^G$ through a linear layer, followed by a 1D causal convolution layer with SiLU activation $\sigma$. This captures local dependencies between continuous features within GPS perspectives to complement the GPS-SSM, producing the input $\boldsymbol{X}_i^G \in \mathbb{R}^{n \times E}$. We then implement the GPS-SSM selection mechanism by constructing input-dependent parameters. Specifically, we randomly initialize the hidden state mapping matrix $\boldsymbol{A} \in \mathbb{R}^H$ as a learnable parameter of the block. We then use the high-order movement behavior feature sequence $\boldsymbol{S}_{\mathcal{T}}$ to calculate the other three parameter matrices $\boldsymbol{B}, \boldsymbol{C}$, and $\boldsymbol{\Delta}$ as follows:

$$\boldsymbol{B} = \mathrm{Linear}(\boldsymbol{S}_{\mathcal{T}}), \ \boldsymbol{C} = \mathrm{Linear}(\boldsymbol{S}_{\mathcal{T}}), \ \boldsymbol{\Delta} = \sigma_{\Delta}(\mathrm{Linear}(\boldsymbol{S}_{\mathcal{T}}) + \boldsymbol{b}_{\Delta}), \quad (1)$$

where Linear represents linear projection, $\sigma_{\Delta}$ denotes the Softplus activation, and $\boldsymbol{b}_{\Delta}$ is the bias parameter of $\boldsymbol{\Delta}$. Here, $\boldsymbol{B}, \boldsymbol{C} \in \mathbb{R}^{n \times N}, \boldsymbol{\Delta} \in \mathbb{R}^{n \times H}$, where $N$ is the state dimension of GPS-SSM, and $H$ is the number of heads. This parameterization extends the state space of GPS-SSM to incorporate high-order movement behaviors, enabling precise control over how trajectory movement behavior changes affect the selective processing of GPS input embeddings. For simplicity, we denote Eq. 1 as: $\boldsymbol{B}, \boldsymbol{C}, \boldsymbol{\Delta} = \mathrm{Param}(\boldsymbol{S}_{\mathcal{T}})$. After parameterization, we process $\boldsymbol{X}_i^G$ through GPS-SSM to obtain the output vector $\boldsymbol{Y}_i^G \in \mathbb{R}^{n \times E}$. A more detailed introduction to SSMs is provided in Appendix A.

The **Road-SSM branch** follows an analogous process. The input $\boldsymbol{Z}_{i-1}^R$ is processed through a linear layer with SiLU activation to obtain $\boldsymbol{X}_i^R \in \mathbb{R}^{n \times E}$. The state space of Road-SSM is extended to trajectory geographical space to capture how changes in trajectory movement details affect road input embeddings selection, implemented as $\boldsymbol{B}', \boldsymbol{C}', \boldsymbol{\Delta}' = \mathrm{Param}(\boldsymbol{Y}_i^G)$. The Road-SSM output vector $\boldsymbol{Y}_i^R \in \mathbb{R}^{n \times E}$ is computed using the same methodology as GPS-SSM.

Finally, we obtain the output embeddings $\boldsymbol{Z}_i^G, \boldsymbol{Z}_i^R$ of the $i$-th Traj-Mamba block as follows:

$$\boldsymbol{Z}_i^G = \mathrm{Linear}(\mathrm{RMSNorm}(\boldsymbol{Y}_i^G \odot \boldsymbol{X}_i^R)), \quad \boldsymbol{Z}_i^R = \mathrm{Linear}(\boldsymbol{Y}_i^R) \qquad (2)$$

Here, the dot-product gating mechanism with $\boldsymbol{X}_i^R$ enables the feature selection of $\boldsymbol{Y}_i^G$ at the road perspective level. $\boldsymbol{Z}_i^G, \boldsymbol{Z}_i^R$ maintain the same shape as $\boldsymbol{Z}_{i-1}^G, \boldsymbol{Z}_{i-1}^R$, and subsequently become the inputs to the next block. After obtaining the outputs $\boldsymbol{Z}_L^G, \boldsymbol{Z}_L^R$ of the last block, we concatenate them and apply mean pooling to derive the trajectory embedding $\boldsymbol{z}_{\mathcal{T}} \in \mathbb{R}^E$ as the final output of the Traj-Mamba encoder.

## 4.2 Travel Purpose-aware Pre-training

To extract travel purpose without adding extra computational load to Traj-Mamba encoder, we propose a Travel Purpose-aware Pre-training scheme. As shown in Fig. 2a, this approach first models both road and POI views of a trajectory using two textual encoders to extract travel purpose, then aligns these representations with Traj-Mamba encoder's output embeddings through contrastive learning.

**Road and POI Views.** Given a trajectory $\mathcal{T} = \langle \tau_1, \tau_2, \ldots, \tau_n \rangle$ and the road network $\mathcal{G}$, we first identify the closest POI $p_i$ to each trajectory point $\tau_i$ based on their geographical distance. Next, we obtain the initial textual embeddings $\boldsymbol{z}_{e_i}, \boldsymbol{z}_{p_i} \in \mathbb{R}^E$ for each road segment $e_i$ and POI $p_i$ using a shared pre-trained textual embedding module $\boldsymbol{E}_{\mathrm{text}}$. To further capture the semantic information of roads and POIs, we consider the local information from neighboring road segments and POIs as well as the global information from the origin $\tau_1$ and destination $\tau_n$ of $\mathcal{T}$, then aggregate them to update the textual embeddings $\boldsymbol{z}_{e_i}$ and $\boldsymbol{z}_{p_i}$. For each unique POI, we also use an index-fetching embedding layer $\boldsymbol{E}_{\mathrm{P_{id}}}$ to assign it a learnable index embedding. The above process can be expressed as:

$$\begin{aligned}
\tilde{\boldsymbol{z}}_{e_i} &= \boldsymbol{z}_{e_i} + \mathrm{Agg}^{\mathrm{Road}}(\boldsymbol{z}_{e_j}, \boldsymbol{z}_{e_1}, \boldsymbol{z}_{e_n} | j \in \mathcal{N}_i) \\
\tilde{\boldsymbol{z}}_{p_i} &= \boldsymbol{z}_{p_i} + \mathrm{Agg}^{\mathrm{POI}}(\boldsymbol{z}_{p_j}, \boldsymbol{z}_{p_1}, \boldsymbol{z}_{p_n} | j \in \mathcal{N}_i') + \boldsymbol{E}_{\mathrm{P_{id}}}(p_i),
\end{aligned} \qquad (3)$$

where $\mathrm{Agg}^{\mathrm{Road}}(\cdot), \mathrm{Agg}^{\mathrm{POI}}(\cdot)$ are aggregation functions, and $\mathcal{N}_i, \mathcal{N}_i'$ denote the sets of neighboring nodes for $e_i$ and $p_i$, respectively. Here we use the residual connection to preserve the raw textual information. More details for road/POI textual embeddings are shown in Appendix I. Finally, the embedding sequences $\langle \tilde{\boldsymbol{z}}_{e_1}, \tilde{\boldsymbol{z}}_{e_2}, \ldots, \tilde{\boldsymbol{z}}_{e_n} \rangle$ and $\langle \tilde{\boldsymbol{z}}_{p_1}, \tilde{\boldsymbol{z}}_{p_2}, \ldots, \tilde{\boldsymbol{z}}_{p_n} \rangle$ are processed through a pair of 2-layer Mamba2 blocks followed by mean pooling, generating the road and POI views $\boldsymbol{z}_{\mathcal{T}}^{\mathrm{Road}}, \boldsymbol{z}_{\mathcal{T}}^{\mathrm{POI}} \in \mathbb{R}^E$ of $\mathcal{T}$.

**Contrastive Learning.** After representing travel purposes as road and POI views, we align the output embeddings from Traj-Mamba encoder with these two views using contrastive learning. Given a trajectory batch $\mathbb{T} = \{\mathcal{T}_i\}_{i=1}^B$ of size $B$, we can obtain their embedding vectors from Traj-Mamba encoder as $\{\boldsymbol{z}_{\mathcal{T}_i}\}_{i=1}^B$. Their road and POI views, as described above, are $\{\boldsymbol{z}_{\mathcal{T}_i}^{\mathrm{Road}}\}_{i=1}^B$ and $\{\boldsymbol{z}_{\mathcal{T}_i}^{\mathrm{POI}}\}_{i=1}^B$. For each trajectory $\mathcal{T}_i$, we regard $\boldsymbol{z}_{\mathcal{T}_i}$ as the anchor, its road and POI views $\boldsymbol{z}_{\mathcal{T}_i}^{\mathrm{Road}}, \boldsymbol{z}_{\mathcal{T}_i}^{\mathrm{POI}}$ as the positive samples, and $\boldsymbol{z}_{\mathcal{T}_j}^{\mathrm{Road}}, \boldsymbol{z}_{\mathcal{T}_j}^{\mathrm{POI}}$ of other trajectories within the batch as the negative samples. Next, we apply the InfoNCE loss [30], denoted by $\mathcal{L}_{\mathbb{T}}^{\mathrm{Road}}, \mathcal{L}_{\mathbb{T}}^{\mathrm{POI}}$. Note that the temperature parameter $T$ is directly optimized during training as a log-parameterized multiplicative scalar [32]. The overall loss is a combination of the two losses, defined as $\mathcal{L}_{\mathbb{T}} = \frac{1}{2}(\mathcal{L}_{\mathbb{T}}^{\mathrm{Road}} + \mathcal{L}_{\mathbb{T}}^{\mathrm{POI}})$.

## 4.3 Knowledge Distillation Pre-training

To further reduce redundancy in trajectories, we employ a Knowledge Distillation Pre-training strategy. As shown in Fig. 2b, we first apply a learnable mask generator for trajectory compression based on the underlying correlations of trajectory features. Next, we designate the travel purpose-aware pre-trained Traj-Mamba encoder as the teacher model and fix its weights, then load these weights to initialize a new encoder to generate embeddings for compressed trajectories. Subsequently, we align

the compressed representation with the full-trajectory embedding from the teacher model to ensure the effectiveness of compression.

**Mask Generator.** Given a trajectory $\mathcal{T} = \langle \tau_1, \tau_2, \ldots, \tau_n \rangle$, we first filter out explicit redundant points, including all intermediate points during vehicle stops and steady pace points on the same road segment, through preprocessing. Next, we input the preprocessed trajectory $\tilde{\mathcal{T}}^{\mathrm{pre}}$ with length $n' \leq n$ into the learnable mask generator to remove the implicit redundancy using the derived mask $\boldsymbol{m}$. Due to the heavy-tailedness of logical masks [39], we apply a sparse stochastic gate $g(\cdot)$ with an intrinsic binary-skewed parameter $\boldsymbol{\mu} \in \mathbb{R}^{n'}$ to maintain feature selection sparsity and reduce the variance in the masks. Specifically, for each point $\tau_i^{\mathrm{pre}}$ of $\tilde{\mathcal{T}}^{\mathrm{pre}}$, the point-specific mask $\boldsymbol{m}_i$ is calculated by $\boldsymbol{m}_i = g(\boldsymbol{\mu}_i) = \max(0, \min(1, \boldsymbol{\mu}_i + \epsilon))$, where $\epsilon \sim \mathcal{N}(0, \delta^2)$ is random noise injected into each point during training and removed during testing, with $\delta$ fixed throughout training. Considering the underlying temporal correlations of trajectory features, we extend the SiLU activation by using these correlations as temperature scaling in the Sigmoid function to construct the smooth $\boldsymbol{\mu}$ as follows:

$$\boldsymbol{\mu} = \mathrm{MeanPool}(\hat{\boldsymbol{\mu}} \odot \mathrm{Sigmoid}(\mathrm{Mamba}(\tilde{\mathcal{T}}^{\mathrm{pre}})\hat{\boldsymbol{\mu}})), \tag{4}$$

where $\hat{\boldsymbol{\mu}}$ is a randomly initialized learnable parameter. A lightweight Mamba block $\mathrm{Mamba}$ is used to efficiently capture the correlations of trajectory features. Finally, we filter $\tilde{\mathcal{T}}^{\mathrm{pre}}$ based on $\boldsymbol{m}$ to obtain the compressed trajectory $\tilde{\mathcal{T}}$. Subsequently, we feed $\tilde{\mathcal{T}}$ into the Traj-Mamba encoder and derive its compressed representation $\tilde{z}_{\mathcal{T}}$ as the trajectory embedding for various downstream tasks.

**Pre-training Loss.** Given a trajectory batch $\mathbb{T} = \{\mathcal{T}_i\}_{i=1}^B$ of size $B$, we can obtain their compressed embeddings as $\tilde{\boldsymbol{Z}} = \{\tilde{z}_{\mathcal{T}_i}\}_{i=1}^B$ and their full-trajectory embeddings from the teacher model as $\boldsymbol{Z} = \{z_{\mathcal{T}_i}\}_{i=1}^B$. Next, we align these embeddings through two optimization objectives. To maximize the information from $\boldsymbol{Z}$ contained in $\tilde{\boldsymbol{Z}}$, we apply the MEC loss [25] following the infomax principle:

$$\mathcal{L}_{\mathbb{T}}^{\mathrm{MEC}} = -\mathrm{trace}\left(\frac{B+E}{2} \sum_{k=1}^{K} \frac{(-1)^{k+1}}{k} \left(\frac{E}{B\varepsilon^2} \boldsymbol{Z}^{\top} \tilde{\boldsymbol{Z}}\right)\right), \tag{5}$$

where $K$ is the order of the Taylor expansion and $\varepsilon$ is the upper bound of the decoding error. This loss maximizes the entropy of embeddings $\tilde{\boldsymbol{Z}}$ while satisfying view consistency between $\tilde{\boldsymbol{Z}}$ and $\boldsymbol{Z}$ (both derived from $\mathbb{T}$), enhancing the generalization of the compressed representations across various downstream tasks. Additionally, we optimize the mask using the Gaussian error function (erf) to constrain the length of compressed trajectories, ensuring encoding efficiency. Formally:

$$\mathcal{L}_{\mathbb{T}}^{\mathrm{mask}} = \mathrm{Mean}\left(\frac{1}{2} + \frac{1}{2}\mathrm{erf}\left(\frac{\langle \boldsymbol{\mu}_{\mathcal{T}_1}, \boldsymbol{\mu}_{\mathcal{T}_2}, \cdots, \boldsymbol{\mu}_{\mathcal{T}_B} \rangle}{\sqrt{2}\delta}\right)\right), \tag{6}$$

where $\boldsymbol{\mu}_{\mathcal{T}_i}$ is the mask generator's parameter for $\mathcal{T}_i$ calculated by Eq. 4. Finally, the overall loss is a balanced combination of these two objectives, denoted as $\mathcal{L}_{\mathbb{T}}' = \frac{1}{2}(\mathcal{L}_{\mathbb{T}}^{\mathrm{MEC}} + \mathcal{L}_{\mathbb{T}}^{\mathrm{mask}})$.

## 5 Experiments

To evaluate the performance of our proposed model, we conduct extensive experiments on two real-world vehicle trajectory datasets, targeting three different types of downstream tasks: Destination Prediction (DP), Arrival time estimation (ATE), and Similar Trajectory Search (STS). The code of TrajMamba is available at *https://github.com/yichenliuzong/TrajMamba*.

**Datasets:** In our experiments, we use two real-world datasets released by Didi[3], called **Chengdu** and **Xian**. Trajectories shorter than 5 or longer than 120 points are excluded from our study, and their points have variations in sampling intervals. For the two datasets, we fetch the information of POIs from the AMap API[4] and road networks from OpenStreetMap[5]. Appendix C provides statistical information for each processed dataset. We split the datasets into training, validation, and testing sets in an 8:1:1 ratio, with departure times in chronological order.

**Baselines:** We compare the proposed model with nine state-of-the-art (SOTA) trajectory learning methods including t2vec [20], Trembr [12], CTLE [23], Toast [6], TrajCL [2], LightPath [41], START [18], MMTEC [22], and JGRM [27]. More details are in Appendix D.

---

[3]https://gaia.didichuxing.com/
[4]https://lbs.amap.com/api/javascript-api-v2
[5]https://www.openstreetmap.org/

Table 1: Destination prediction (DP) performance results. ↑: higher better, ↓: lower better. **Bold**: the best, underline: the second best. The results under **w/o ft** setting can be found in Appendix E.

| Dataset | Chengdu | | | | | Xian | | | | |
|---|---|---|---|---|---|---|---|---|---|---|
| | GPS | | Road Segment | | | GPS | | Road Segment | | |
| Method | RMSE ↓ (meters) | MAE ↓ (meters) | Acc@1 ↑ (%) | Acc@5 ↑ (%) | Recall ↑ (%) | RMSE ↓ (meters) | MAE ↓ (meters) | Acc@1 ↑ (%) | Acc@5 ↑ (%) | Recall ↑ (%) |
| t2vec | 579.30 | 387.50 | 47.74 | 73.51 | 16.64 | 482.64 | 310.08 | 43.60 | 74.67 | 13.53 |
| Trembr | 505.62 | 376.88 | 48.99 | 72.08 | 17.01 | 473.97 | 301.45 | 44.50 | 75.11 | 12.90 |
| CTLE | 430.19 | 382.82 | 51.00 | 79.43 | 21.47 | 477.70 | 384.08 | 44.84 | 76.78 | 14.83 |
| Toast | 480.52 | 412.58 | 50.90 | 79.66 | 21.07 | 523.76 | 443.99 | 45.08 | 77.65 | 15.46 |
| TrajCL | 365.50 | 272.63 | 50.85 | 79.69 | 21.57 | 383.39 | 262.20 | 45.81 | 79.06 | 16.84 |
| LightPath | 553.27 | 360.86 | 49.15 | 78.59 | 20.66 | 598.20 | 348.61 | 44.39 | 72.75 | 14.42 |
| START | 333.10 | 240.40 | 52.78 | 80.42 | 23.32 | 319.00 | 208.35 | 46.13 | 79.34 | 16.31 |
| MMTEC | 312.78 | 212.78 | 52.85 | 79.55 | 22.69 | 311.99 | 207.94 | 47.45 | 79.45 | 17.90 |
| JGRM | 215.99 | 172.79 | 54.27 | 85.06 | 25.68 | 303.12 | 226.82 | 48.21 | 81.79 | 19.01 |
| **TrajMmaba** | **129.47** | **85.95** | **58.21** | **86.81** | **30.45** | **236.33** | **155.63** | **48.81** | **83.37** | **20.55** |

**Implement Details:** The four key hyper-parameters of TrajMamba and their optimal values are $L = 5$, $E = 256$, $N = 32$, and $H = 4$. Their effectiveness is reported in the subsequent section. Both travel purpose-aware pre-training and knowledge distillation pre-training schemes perform 15 epochs on the training set. More detailed experimental setup is shown in Appendix B. We run each set of experiments 5 times and report their mean values of the metrics.

### 5.1 Destination Prediction

**Setups:** DP task involves predicting the destination of a trajectory at both GPS and road segment levels. When calculating the embedding $z_{\mathcal{T}}$ of a trajectory $\mathcal{T}$, its last 5 points are omitted. A fully connected network then uses this embedding to predict the destination's GPS coordinates or road segment. For GPS prediction, Mean Squared Error (MSE) is used as the loss function, while Mean Absolute Error (MAE) and Root Mean Squared Error (RMSE) of the shortest distance on the Earth's surface serve as evaluation metrics. For road segment prediction, we use the cross-entropy loss, with Accuracy@N (Acc@1, Acc@5) and macro-Recall as evaluation metrics. For this task, we can either fine-tune trajectory learning methods with task supervision (default) or fix their parameters and only update the predictors parameters, denoted as *without fine-tune* (**w/o ft**).

**Results:** Our brief results are shown in Tab. 1, and consistently surpass all baselines. The comparison with JGRM, the second best model, is particularly noteworthy. For road segment prediction, our model achieves average performance gains of **9.30%** and **3.75%** over JGRM on two datasets, respectively. And for GPS prediction, our improvements are even more significant, exceeding **45.16%** and **26.71%**. Our model performs exceptionally well in the DP task thanks to its effective extraction of both movement patterns and travel purposes, allowing for precise identification of trajectory destinations.

### 5.2 Arrival Time Estimation

**Setups:** ATE task aims to predict the arrival time of a trajectory still in movement, corresponding to real-world scenarios such as ride-hailing arrival time estimation and traffic management. Similar to the DP task, we omit the last 5 points of a trajectory $\mathcal{T}$, use a fully connected network for prediction, and adopt either fine-tune (default) or **w/o ft** setting. MSE supervises the prediction, while MAE, RMSE, and Mean Absolute Percentage Error (MAPE) are used as evaluation metrics.

**Results:** The performance results of this task are shown in Tab. 2, and our model consistently surpasses all baselines. Under fine-tune setting, TrajMamba remains competitive even when compared with the SOTA model, JGRM, by **32.35%** and **23.44%** on two datasets. Moreover, our model shows superior performance under **w/o ft** setting, which further highlights the effectiveness of its pre-training process and sufficient modeling of continuous features.

### 5.3 Similar Trajectory Search

**Setups:** STS task aims to identify the most similar trajectory to a query trajectory from a batch of candidates. Similarities between trajectories are calculated with the cosine similarity between their embeddings. Accuracy@N (Acc@1, Acc@5) and Mean Rank are used as evaluation metrics. The

Table 2: Arrival time estimation (ATE) performance results. A lower value indicates better performance. **Bold**: the best, underline: the second best.

| Strategy | fine-tune / without fine-tune | | | | | |
|---|---|---|---|---|---|---|
| Dataset | Chengdu | | | Xian | | |
| Method \ Metric | RMSE (seconds) | MAE (seconds) | MAPE (%) | RMSE (seconds) | MAE (seconds) | MAPE (%) |
| t2vec | 127.41 / 138.30 | 64.67 / 79.74 | 14.01 / 18.71 | 214.40 / 207.11 | 108.80 / 117.86 | 16.96 / 16.01 |
| Trembr | 124.32 / 159.60 | 63.42 / 110.36 | 13.60 / 29.50 | 209.12 / 435.04 | 107.02 / 337.35 | 16.40 / 47.11 |
| CTLE | 135.21 / 135.59 | 55.41 / 63.45 | 11.18 / 13.99 | 207.16 / 272.88 | 107.46 / 176.16 | 16.25 / 31.84 |
| Toast | 171.58 / 149.67 | 91.66 / 79.69 | 18.84 / 17.89 | 202.99 / 299.94 | 102.73 / 205.49 | 15.75 / 32.55 |
| TrajCL | 132.98 / 136.56 | 55.78 / 79.59 | 11.86 / 19.85 | 183.74 / 194.64 | 73.21 / 106.66 | 12.55 / 16.80 |
| LightPath | 123.00 / 129.48 | 58.04 / 56.82 | 12.83 / 12.71 | 169.01 / 186.02 | 74.08 / 77.33 | 10.50 / 10.41 |
| START | 121.11 / 144.54 | 58.97 / 79.78 | 13.49 / 19.72 | 159.89 / 213.22 | 72.19 / 120.74 | 10.26 / 20.01 |
| MMTEC | 105.72 / 124.98 | 40.48 / 54.76 | 8.17 / 12.13 | 155.59 / 183.66 | 58.84 / 78.14 | 7.65 / 12.03 |
| JGRM | 65.24 / 119.40 | 31.52 / 82.02 | 6.84 / 22.01 | 84.49 / 202.81 | 39.92 / 137.75 | 5.97 / 20.20 |
| **TrajMmaba** | **50.17 / 92.52** | **17.73 / 44.89** | **4.77 / 11.07** | **74.75 / 134.57** | **25.99 / 55.74** | **4.55 / 8.79** |

Table 3: Similar trajectory search (STS) performance results. ↑: higher better, ↓: lower better. **Bold**: the best, underline: the second best.

| Dataset | Chengdu / Xian | | |
|---|---|---|---|
| Method \ Metric | Acc@1 ↑ (%) | Acc@5 ↑ (%) | Mean ↓ Rank |
| t2vec | 75.20 / 89.83 | 88.53 / 94.87 | 5.04 / 5.71 |
| Trembr | 84.60 / 78.75 | 89.63 / 86.45 | 10.01 / 11.04 |
| CTLE | 69.07 / 46.80 | 78.50 / 58.15 | 41.69 / 48.32 |
| Toast | 71.73 / 32.60 | 81.70 / 60.00 | 46.16 / 46.02 |
| TrajCL | 94.45 / 90.87 | 96.70 / 94.50 | 2.88 / 4.86 |
| LightPath | 68.35 / 74.50 | 82.05 / 88.75 | 17.94 / 6.99 |
| START | 91.07 / 91.43 | 96.57 / 96.23 | 4.90 / 4.26 |
| MMTEC | 90.17 / 82.97 | 96.47 / 91.17 | 3.76 / 4.86 |
| JGRM | 86.63 / 86.97 | 96.10 / 96.77 | 1.97 / 1.62 |
| **TrajMmaba** | **96.53 / 97.97** | **99.13 / 99.80** | **1.15 / 1.07** |

Table 4: Efficiency analysis of methods. A lower value indicates better performance. **Bold**: the best, underline: the second best.

| Dataset | Chengdu / Xian | | |
|---|---|---|---|
| Method \ Metric | Model size (MBytes) | Train time (min/epoch) | Embed time (seconds) |
| t2vec | **1.641/1.415** | 2.783/5.937 | 4.722/10.613 |
| Trembr | 5.752/5.301 | 3.360/6.067 | 3.527/10.351 |
| CTLE | 3.756/3.756 | 4.533/14.354 | 16.736/38.318 |
| Toast | 4.008/3.557 | 4.400/10.650 | 13.812/38.056 |
| TrajCL | 4.382/3.932 | 7.699/14.567 | 9.998/22.838 |
| LightPath | 12.958/12.507 | 10.250/23.217 | 25.726/61.404 |
| START | 15.928/15.026 | 15.927/37.528 | 29.034/67.801 |
| MMTEC | 1.888/1.663 | **0.378/0.753** | 27.068/40.143 |
| JGRM | 23.088/20.827 | 22.339/62.791 | 94.048/251.898 |
| **TrajMamba** | 5.407/5.182 | 3.225/7.273 | **1.721/3.918** |

parameters of trajectory learning methods are fixed after pre-training. Since most datasets don't have labeled data for this task, we create labels following the method introduced in Appendix B.2.

**Results:** Our brief results are shown in Tab. 3. TrajMambas best performance demonstrates that its pretraining process effectively extracts travel semantics and reduces redundancy from trajectories, while baselines underperform due to the inadequate modeling of these aspects. Notably, CTLE and Toast do not learn a trajectory-level representation directly, resulting in poor performance.

## 5.4 Model Efficiency

Tab. 4 compares the efficiency of different methods on both datasets. In terms of model size and embed time, TrajMamba demonstrates high computational efficiency, achieving the same lightweight and embed speed as RNN-based methods such as TremBR and t2vec. It is significantly more efficient compared to Transformer-based methods like START and LightPath. Notably, TrajMamba shows a strong advantage in all three terms compared to JGRM, which is the closest to it in terms of effect. Appendix F provides detailed computation cost analysis of TrajMamba. Given TrajMamba's superior performance across a variety of tasks, it achieves its design goal of semantic-rich trajectory learning with high efficiency. It is worth noting that TrajMamba does not have a particularly short training time. However, since the pre-training process does not add extra burden to the embedding process, where efficiency is more critical in real-world applications, the additional training time can be considered worthwhile due to its effectiveness.

## 5.5 Model Analysis

**Ablation Study.** As shown in Tab. 5, we have the following observations: 1) Different modules focus on different types of tasks, and the overall performance of our method beats all variants. 2) *V-Mamba* and *V-Transformer* show performance degradation, proving the contribution of the Traj-Mamba blocks. 3) *w/o Purpose* and *w/o KD* all have worse performance compared to the complete model, showing that both pre-training procedures contribute to TrajMambas performance. 4) The worse

Table 5: Ablations on Chengdu dataset in all tasks. ↑: higher better, ↓: lower better. **Bold**: the best, underline: the second best. The setting of variants can be found in Appendix G.

| Task | Destination Prediction | | | | | Arrival Time Estimation | | | Similar Trajectory Search | | |
|---|---|---|---|---|---|---|---|---|---|---|---|
| | GPS | | Road Segment | | | | | | | | |
| Metric
Method | RMSE ↓
(meters) | MAE ↓
(meters) | Acc@1 ↑
(%) | Acc@5 ↑
(%) | Recall ↑
(%) | RMSE ↓
(seconds) | MAE ↓
(seconds) | MAPE ↓
(%) | Acc@1 ↑
(%) | Acc@5 ↑
(%) | Mean ↓
Rank |
| V-Mamba | 152.51 | 109.84 | 56.93 | 85.61 | 29.36 | 51.25 | 20.22 | 5.62 | 94.63 | 98.00 | 1.36 |
| V-Transformer | 182.00 | 139.76 | 57.56 | 86.44 | 29.31 | 56.59 | 19.84 | 5.23 | 93.90 | 97.45 | 1.41 |
| w/o Purpose | 153.84 | 116.82 | 58.07 | 86.25 | 29.94 | 53.21 | 18.78 | 5.30 | 85.35 | 91.95 | 5.83 |
| w/o KD | 145.34 | 106.60 | 52.83 | 80.74 | 22.48 | 53.26 | 19.52 | 5.35 | 95.40 | 99.10 | 1.65 |
| w/o Compress | 174.65 | 129.93 | 48.88 | 76.94 | 18.90 | 51.77 | 21.37 | 5.13 | 94.93 | 98.17 | 3.88 |
| MG-Trans-DP | 173.95 | 129.12 | 48.83 | 77.37 | 18.93 | 50.55 | 20.30 | 5.34 | 96.10 | 98.80 | 1.38 |
| MG-trans-DS | 150.70 | 106.35 | 53.12 | 80.46 | 23.16 | 54.70 | 19.55 | 5.56 | 95.10 | 98.80 | 2.16 |
| w/o Filter | **125.56** | 86.49 | **58.31** | 86.75 | **30.52** | 50.18 | **17.41** | **4.56** | **96.90** | 99.00 | 1.22 |
| **TrajMmaba** | 129.47 | **85.95** | 58.21 | **86.81** | 30.45 | **50.17** | 17.73 | 4.77 | 96.53 | **99.13** | **1.15** |

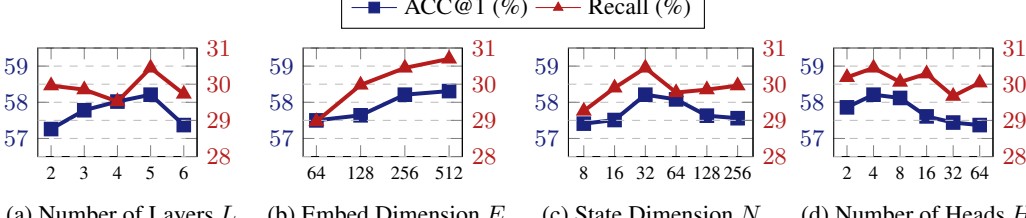

(a) Number of Layers $L$    (b) Embed Dimension $E$    (c) State Dimension $N$    (d) Number of Heads $H$

Figure 4: Effectiveness of hyper-parameters validated on Chengdu dataset.

performance witnessed by *w/o Compress* demonstrates that reducing redundancy in trajectories contributes to embedding quality. 5) *MG-Trans-DP* performs well in STS but poorly in DP, indicating the advantages of learnable trajectory compression for various downstream tasks; while the results of *MG-Trans-DS* verify our mask generator outperforms direct downsampling, highlighting its ability to identify key trajectory points. 6) *w/o Filter* performs comparably to the full model, validating the mask generator's effectiveness lies in its learnable soft masking mechanism rather than filtering preprocessing.

**Hyperparameter Analysis.** We analyze the effectiveness of hyper-parameters $L$, $E$, $N$, and $H$ of TrajMamba based on the Acc@1 and Recall of the DP task on **Chengdu**s validation set. As shown in Fig. 4a-c, $L$, $E$, and $N$ control the model capacity, where $E$ has the most prominent effect since it directly controls the dimension of the final trajectory embeddings. Fig. 4b shows that increasing $E$ from 256 to 512 not only yields limited performance improvement but also increases model size and encoding time, conflicting with our goal of efficient inference. After balancing performance and efficiency, their optimal values are $L = 5$, $E = 256$, and $N = 32$. In Fig. 4d, $H$ determines the complexity of the multi-input SSMs in Traj-Mamba encoder, with an optimal value of 4.

**Scalability.** We analyze the scalability of the proposed model against JGRM, one of the SOTA models, on Chengdu dataset. As shown in Fig. 5a, we use varying proportions of the training data: 100%, 40%, and 20% for the DP task and report the valid Acc@1 on several fine-tune epochs. It can be seen that our model demonstrates faster progress and achieves superior performance with less data compared to JGRM. In Fig. 5b, the ATE results of models fine-tuned on dataset of different sizes also confirms this. This shows that our model can be adapted to downstream tasks with lightweight fine-tuning, which is attributed to our well-designed pre-training process. And considering the high encoding efficiency of our model, it have good practicality in real-world applications of trajectories.

**Case Study.** We analyze the effectiveness of TrajMamba's learnable trajectory compression. As shown in Fig. 6a, there are many redundant points in trajectory due to its high sampling frequency. After rule-based filtering, explicit redundant points are removed, yet the resulting trajectory shown in Fig. 6b still retains many non-critical points that can be considered implicit redundant. Fig. 6c shows that the pre-trained mask generator accurately locates key trajectory points closely linked to revealing travel semantics and eliminates the implicit redundancy, thereby outputting the effective compressed trajectory to achieve high encoding efficiency and embedding quality. Notably, trajectory points near the origin and destination are largely preserved, as they carry semantics strongly associated with travel purpose.

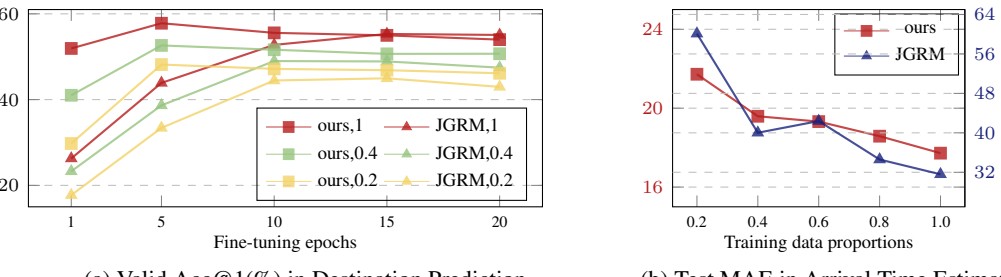

(a) Valid Acc@1(%) in Destination Prediction          (b) Test MAE in Arrival Time Estimation

Figure 5: Scalability of fine-tuning on Chengdu dataset.

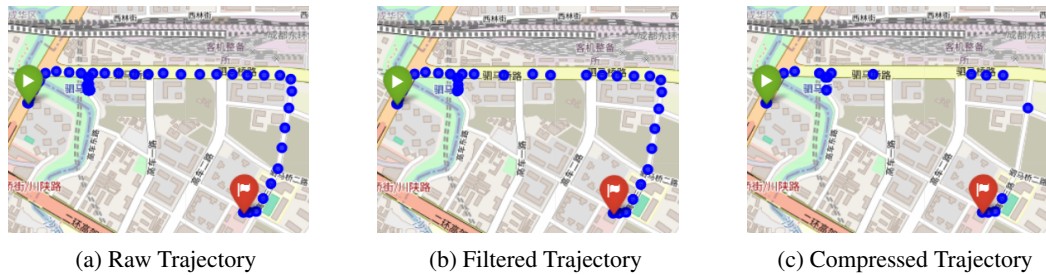

(a) Raw Trajectory                    (b) Filtered Trajectory                    (c) Compressed Trajectory

Figure 6: Case Study on Chengdu dataset.

**Quantitative Study on Knowledge Distillation Pre-training Loss Weights.** We analyze how different weights of $\mathcal{L}_{\mathbb{T}}^{\mathrm{MEC}}$ and $\mathcal{L}_{\mathbb{T}}^{\mathrm{mask}}$ affect downstream tasks and compression preformance on Chengdu dataset. Larger $\mathcal{L}_{\mathbb{T}}^{\mathrm{MEC}}$ weight masks compressed representations containing more information from full-trajectory embeddings; Larger $\mathcal{L}_{\mathbb{T}}^{\mathrm{mask}}$ weight yields shorter compressed trajectories from the mask generator. As shown in Tab. 6, weights (0.5, 0.5) help the model balance these two objectives, achieving the best comprehensive downstream performance and efficient encoding time.

Table 6: Impact of different weights for the loss functions $\mathcal{L}_{\mathbb{T}}^{\mathrm{MEC}}$ and $\mathcal{L}_{\mathbb{T}}^{\mathrm{mask}}$ on Chengdu dataset in all tasks. ↑: higher better, ↓: lower better. **Bold**: the best, underline: the second best.

| Task | Destination Prediction | | | | | Arrival Time Estimation | | | Similar Trajectory Search | | | Efficiency |
|---|---|---|---|---|---|---|---|---|---|---|---|---|
| Metric | GPS | | Road Segment | | | RMSE↓ (seconds) | MAE↓ (seconds) | MAPE↓ (%) | Acc@1↑ (%) | Acc@5↑ (%) | Mean↓ Rank | Embed Time↓ (seconds) |
| Method | RMSE↓ (meters) | MAE↓ (meters) | Acc@1↑ (%) | Acc@5↑ (%) | Recall↑ (%) | | | | | | | |
| (1.0, 0.0) | **125.62** | 86.54 | 56.28 | 83.85 | 26.36 | 51.77 | 19.66 | 5.34 | 95.80 | 98.30 | 1.66 | 1.944 |
| (0.7, 0.3) | 135.45 | 91.97 | 58.13 | 86.36 | 30.08 | 54.51 | 17.94 | 4.80 | **96.60** | 98.90 | 1.16 | 1.731 |
| (0.3, 0.7) | 134.87 | 90.07 | 58.15 | 86.55 | 30.03 | 50.98 | 18.56 | 5.05 | 95.80 | 98.80 | 1.20 | **1.692** |
| **(0.5, 0.5) (Ours)** | 129.47 | **85.95** | **58.21** | **86.81** | **30.45** | **50.17** | **17.73** | **4.77** | 96.53 | **99.13** | **1.15** | 1.721 |

## 6 Conclusion

We propose TrajMamba, a new method for efficient and semantic-rich trajectory learning. First, a Traj-Mamba Encoder is proposed to capture movement patterns by jointly modeling GPS and road perspectives. Next, a Travel Purpose-aware Pre-training procedure is introduced to help TrajMamba extract travel purposes from trajectories while maintaining its efficiency. Third, a Knowledge Distillation Pre-training is designed for key trajectory point identification through a learnable mask generator and deriving effective compressed representations. Finally, extensive experiments on two real-world vehicle trajectory datasets and three representative tasks demonstrate the effectiveness and efficiency of TrajMamba. Our work enables a more accurate and efficient analysis of vehicle trajectories, benefiting human mobility services and urban traffic management.

**Limitations** Our TrajMamba is hindered by the different travel semantics from road segments and POIs in trajectory data across datasets, causing model cross-city migration and zero-shot experiments unfeasible. Future work will focus on developing universal road and POI embeddings to enhance cross-city migration and improve model transferability.

**Acknowledgment.** This work was supported by the National Natural Science Foundation of China (No. 62372031) and the Beijing Natural Science Foundation (Grant No. 4242029).

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

## Appendix

Here we introduce the background of SSMs in Sec.A. The implement details including experiment settings and label construction for STS task can be found in Sec.B. The information of datasets are shown in Sec.C. An introduction to Baselines is presented in Sec.D. We display the complete results of three downstream tasks in Sec. E, supplementing the standard deviations of each set of experiments. The computation cost analysis of our model's downstream components is shown in Sec. F. The settings of ablation variants are listed in Sec. G. Moreover, we introduce the detailed calculation processes of trajectory spatio-temporal feature embedding and road/POI textual embedding in Sec.H and Sec.I, respectively.

## A  Structured State Space Models

Structured State Space Models (SSMs) are a recent class of sequence models for deep learning that have proven to be effective at handling long-range models [14]. They are inspired by a pair of linear differential equations that maps a 1-dimensional function $x(t) \in \mathbb{R} \mapsto y(t) \in \mathbb{R}$ through an intermediate hidden state $\boldsymbol{h}(t) \in \mathbb{R}^N$, formulated as:

$$\boldsymbol{h}'(t) = \boldsymbol{A}h(t) + \boldsymbol{B}x(t), \quad y(t) = \boldsymbol{C}h(t), \tag{7}$$

where $\boldsymbol{A} \in \mathbb{R}^{N \times N}$, $\boldsymbol{B} \in \mathbb{R}^{N \times 1}$, and $\boldsymbol{C} \in \mathbb{R}^{1 \times N}$ are the hidden state, input and output mapping matrices, respectively. When applying SSM to discrete sequence data, an additional timescale parameter $\boldsymbol{\Delta}$ is introduced to transform the continuous parameters $\boldsymbol{A}, \boldsymbol{B}$ to discrete parameters $\bar{\boldsymbol{A}}, \bar{\boldsymbol{B}}$ through fixed discretization rules. The most common method is zero-order hold (ZOH) defined by $\bar{\boldsymbol{A}} = \exp(\boldsymbol{\Delta}\boldsymbol{A})$ and $\bar{\boldsymbol{B}} = (\boldsymbol{\Delta}\boldsymbol{A})^{-1}(\exp(\boldsymbol{\Delta}\boldsymbol{A}) - \boldsymbol{I}) \cdot \boldsymbol{\Delta}\boldsymbol{B}$. Then the discretized version of Eq. 7 can be obtained by $\bar{\boldsymbol{A}}, \bar{\boldsymbol{B}}$ as follows:

$$h_t = \bar{\boldsymbol{A}}h_{t-1} + \bar{\boldsymbol{B}}x_t, \quad y_t = \boldsymbol{C}h_t \tag{8}$$

Due to the linear-time invariance (LTI) property of Eq. 7, Eq. 8 can be reformulated as a convolution by unrolling the recurrence, enabling efficient parallelized computation during training. For multidimensional input $\boldsymbol{x}_t \in \mathbb{R}^D$, the above dynamics are typically applied to each channel independently. This is actually completely similar to how multi-head attention works, and thus further leads to the concept of multi-head SSM.

**Multi-input SSM** introduced in Mamba2 [7] creates $H$ heads by reshaping the input $\boldsymbol{x}_t \in \mathbb{R}^D$ into $\boldsymbol{x}_t \in \mathbb{R}^{H \times \frac{D}{H}}$. For the $i$-th head, the recurrence formulations of multi-input SSM when $D/H = 1$ are as follows:

$$\boldsymbol{h}_{t,i} = \bar{\boldsymbol{A}}_i \boldsymbol{I} \boldsymbol{h}_{t-1,i} + \bar{\boldsymbol{B}}_i \boldsymbol{x}_{t,i}, \quad \boldsymbol{y}_{t,i} = \boldsymbol{C}\boldsymbol{h}_{t,i}, \tag{9}$$

where $\boldsymbol{x}_{t,i}, \boldsymbol{h}_{t,i}, \bar{\boldsymbol{A}}_i$, and $\bar{\boldsymbol{B}}_i$ denote the $i$-th row of $\boldsymbol{x}_t, \boldsymbol{h}_t, \bar{\boldsymbol{A}}$, and $\bar{\boldsymbol{B}}$, respectively. Here, $\boldsymbol{h}_t \in \mathbb{R}^{H \times N}$, $\bar{\boldsymbol{A}} \in \mathbb{R}^H$, $\bar{\boldsymbol{B}} \in \mathbb{R}^{N \times N}$, and the identity matrix $\boldsymbol{I} \in \mathbb{R}^{N \times N}$. Finally, the output $\boldsymbol{y}_t \in \mathbb{R}^{H \times \frac{D}{H}}$ is reshaped back to $\boldsymbol{y}_t \in \mathbb{R}^D$ by aggregating $H$ heads. When $D/H > 1$, the input $\boldsymbol{x}_{t,i}$ of the $i$-th head can be treated as $D/H$ independent sequences, and then Eq. 9 can be applied to each sequence.

**Selection Mechanisms**  Models using the LTI formulation guarantee computational efficiency at the expense of inability to dynamically focus on specific inputs. To addressing this issue, Mamba [13] introduce the selective SSM, which makes the parameters $\boldsymbol{B}$, $\boldsymbol{C}$, and $\boldsymbol{\Delta}$ depend on the input $\boldsymbol{x} \in \mathbb{R}^{L \times D}$, formulated as:

$$\boldsymbol{B} = \text{Linear}(\boldsymbol{x}), \ \boldsymbol{C} = \text{Linear}(\boldsymbol{x}), \ \boldsymbol{\Delta} = \sigma_\Delta(\text{Linear}(\boldsymbol{x}) + \boldsymbol{b}_\Delta) \tag{10}$$

In this way, each token of the input has its own unique input-dependent parameters, enabling the model to selectively process the input by focusing on or ignoring specific tokens. Then, Mamba employs the hardware-efficient algorithm to ensure linear computational complexity with respect to the input length $n$.

## B  Implement Details

### B.1  Settings

The TrajMamba model is implemented using PyTorch [31]. All models are trained on the training set and evaluated on the testing set. The travel purpose-aware pre-training and the knowledge distillation

pre-training both perform 15 epochs, while the downstream predictors are early-stopped based on the validation set. The final metrics are calculated on the testing set. All experiments are performed five times, and the means and standard deviations are calculated. For model training, we use the Adam optimizer with a batch size of $B = 128$, and the initial learning rates of two pre-training procedures are 0.001 and 0.0001, respectively. We set the radius for neighboring POIs selection $R = 300$ meters. In Eq. 14, we set $\alpha = 1.0, \beta = 0.5$ to balance each factor. For $K$ and $\varepsilon$ in Eq. 5, we follow the same settings in [22]. The experiments are conducted on servers equipped with Intel(R) Xeon(R) W-2155 CPUs and nVidia(R) TITAN RTX GPUs.

## B.2  Label Construction for Similar Trajectory Search

Considering that the datasets we used are recorded by taxis operating in two cities, trajectories sharing the same origin and destination (OD) pair can be regarded as more similar. For each trajectory $\mathcal{T}$ in the test dataset, we first construct its similar trajectory candidate set by collecting all test trajectories that have the same OD pairs as it. The difference between $\mathcal{T}$ and each candidate is quantified by combining their GPS sequence distance calculated by DTW [29] algorithm and the number of differing road segments. In particular, we split $\mathcal{T}$ into two subsequences of odd- and even-numbered points, calculate their difference as the benchmark, and compare it with each candidate. If a candidate's difference from $\mathcal{T}$ is below the benchmark, we use $\mathcal{T}$ as the query $\mathcal{T}^q$ and the most similar candidate as the target $\mathcal{T}^t$. Otherwise, $\mathcal{T}^q$ and $\mathcal{T}^t$ are created from the odd- and even-numbered subsequences of $\mathcal{T}$, respectively. For the STS task, we randomly select 1,000 test trajectories, ensuring that more than 20% are not self-similar to maintain task difficulty. For each query, we exclude trajectories whose OD pairs are within 500 meters away from query's OD pair, and then randomly select 5,000 additional trajectories from the rest of the test dataset to use as the database.

## C  Datasets

The two real-world datasets **Chengdu** and **Xian** consist of vehicle trajectories recorded by taxis operating in Chengdu and Xian, China. Due to the original trajectories having very dense sampling intervals, we retain a portion of the trajectory points through a three-hop resampling process, making most trajectories having sampling intervals of no less than 6 seconds. After resampling, trajectories with fewer than 5 or more than 120 trajectory points are considered anomalies and excluded. Additionally, we retrieve the information of POIs within these datasets' areas of interest from the AMap API, and obtain the road network topology and information from OpenStreetMap. The statistics of these datasets after the above preprocessing are listed in Table 7.

Table 7: Dataset statistics.

| Dataset | Chengdu | Xian |
|---|---|---|
| Time span | 09/30 - 10/10, 2018 | 09/29 - 10/15, 2018 |
| #Trajectories | 140,000 | 210,000 |
| #Points | 18,832,411 | 18,267,440 |
| #Road segments | 4,315 | 3,392 |
| #POIs | 12,439 | 3,900 |

## D  Overview of Baselines

- **t2vec** [20]: Pre-trains the model by reconstructing original trajectories from low-sampling ones using a denoising auto-encoder.

- **Trembr** [12]: Constructs an RNN-based seq2seq model to recover road segments and the time of the input trajectories.

- **CTLE** [23]: Pre-trains a bi-directional Transformer with two MLM tasks for location and hour predictions. The trajectory representation is obtained by mean pooling on point embeddings.

- **Toast** [6]: Uses a context-aware node2vec model to generate segment representations and trains the model with an MLM-based task and a sequence discrimination task.

- **TrajCL** [2]: Introduces a dual-feature self-attention-based encoder and trains the model in a contrastive style using the InfoNCE loss.

- **LightPath** [41]: Constructs a sparse path encoder and trains it with a path reconstruction task and a cross-view & cross-network contrastive task.

- **START** [18]: Includes a time-aware trajectory encoder and a GAT that considers the transitions between road segments. The model is trained with both an MLM task and a contrastive task.

- **MMTEC** [22]: Combines an attention-based discrete encoder and a NeuralCDE-based continuous encoder. The model is trained with a maximum entropy pretext task based on contrastive style.

- **JGRM** [27]: Uses two encoders to separately embed route and GPS trajectories and then employs a modal interactor for information fusion. An MLM task and a cross-modal contrastive task are designed for model training.

## E   Performance Comparison in Three Downstream Tasks

For the DP task, methods can also be trained with fixed parameters. Tab. 8 compares the DP performance of different methods *without fine-tune* on two datasets. We observe that TrajMamba also consistently shows superior performance in this setting, demonstrating that its pretraining process extracts rich information from vehicle trajectories without additional task-specific supervision. For GPS prediction, our model achieves average performance gains of **9.37%** and **9.86%** over the second best model JGRM on two datasets, respectively. And for road segment prediction, our improvements are even more significant, exceeding **27.47%** and **11.65%**.

Table 8: Destination prediction (DP) performance results with standard deviations under **w/o ft** setting. ↑: higher better, ↓: lower better. **Bold**: the best, underline: the second best.

| Dataset | Chengdu | | | | | Xian | | | | |
|---|---|---|---|---|---|---|---|---|---|---|
| | GPS | | Road Segment | | | GPS | | Road Segment | | |
| Method | RMSE ↓ (meters) | MAE ↓ (meters) | Acc@1 ↑ (%) | Acc@5 ↑ (%) | Recall ↑ (%) | RMSE ↓ (meters) | MAE ↓ (meters) | Acc@1 ↑ (%) | Acc@5 ↑ (%) | Recall ↑ (%) |
| t2vec | 2329.63±21.09 | 1868.49±19.49 | 10.77±0.43 | 25.37±0.59 | 1.64±0.11 | 2582.14±46.79 | 2235.27±39.44 | 9.55±0.73 | 22.60±1.16 | 0.88±0.10 |
| Trembr | 1787.18±92.01 | 1419.58±88.95 | 13.45±0.20 | 28.52±0.24 | 1.29±0.03 | 2067.80±196.30 | 1749.76±178.82 | 13.97±4.82 | 27.78±6.21 | 1.04±0.55 |
| CTLE | 3421.09±17.10 | 3041.49±23.49 | 2.39±0.12 | 7.84±0.18 | 0.08±0.00 | 3548.88±4.27 | 3320.46±1.12 | 3.80±0.03 | 10.59±0.05 | 0.06±0.00 |
| Toast | 3434.84±9.55 | 3061.91±14.99 | 2.48±0.15 | 8.55±0.16 | 0.10±0.02 | 3549.65±6.42 | 3325.48±8.21 | 3.77±0.07 | 9.97±0.63 | 0.06±0.01 |
| TrajCL | 1059.81±16.22 | 865.48±10.60 | 30.32±0.31 | 55.51±0.24 | 9.21±0.26 | 1268.41±19.57 | 1054.21±18.54 | 26.97±0.14 | 49.83±0.43 | 5.33±0.12 |
| LightPath | 2365.87±57.52 | 1948.97±57.78 | 19.45±1.08 | 36.11±1.34 | 3.96±0.35 | 2177.37±60.03 | 1859.35±48.50 | 18.55±0.12 | 35.83±0.26 | 2.40±0.10 |
| START | 1347.13±30.72 | 1111.77±29.11 | 25.64±0.20 | 48.42±0.44 | 6.37±0.07 | 1406.06±18.42 | 1173.62±17.18 | 24.11±0.12 | 45.16±0.39 | 4.44±0.18 |
| MMTEC | 2137.14±32.13 | 1687.63±24.08 | 16.44±0.48 | 34.21±0.38 | 2.76±0.26 | 2624.86±21.74 | 2310.00±26.19 | 10.67±0.26 | 24.64±0.18 | 0.94±0.03 |
| JGRM | 406.15±3.64 | 319.06±2.54 | 42.49±0.55 | 73.06±0.61 | 14.93±0.55 | 432.63±14.48 | 339.36±12.87 | 39.57±0.37 | 71.93±0.62 | 12.08±0.24 |
| **TrajMmaba** | **365.60±20.78** | **291.12±16.88** | **51.95±0.62** | **77.72±1.09** | **22.95±0.81** | **386.76±3.88** | **308.42±3.81** | **42.89±0.19** | **74.56±0.29** | **14.85±0.21** |

Additionally, Tab. 1,2 and 3 in Section 5 only report mean values of the metrics in three downstream task. To supplement results in these tables, we report performance results with standard deviations for each task in Tab. 9 ,10 and 11, respectively. It can be seen that our model maintains the best mean values of the metrics in all downstream tasks while having relatively small standard deviations compared to most baseline models. This shows that our model can outcomes robust trajectory embeddings across various tasks.

Table 9: Destination prediction (DP) performance results with standard deviations under fine-tune setting. ↑: higher better, ↓: lower better. **Bold**: the best, underline: the second best.

| Dataset | Chengdu | | | | | Xian | | | | |
|---|---|---|---|---|---|---|---|---|---|---|
| | GPS | | Road Segment | | | GPS | | Road Segment | | |
| Method | RMSE ↓ (meters) | MAE ↓ (meters) | Acc@1 ↑ (%) | Acc@5 ↑ (%) | Recall ↑ (%) | RMSE ↓ (meters) | MAE ↓ (meters) | Acc@1 ↑ (%) | Acc@5 ↑ (%) | Recall ↑ (%) |
| t2vec | 579.30±11.94 | 387.50±4.03 | 47.74±0.24 | 73.51±0.15 | 16.64±0.11 | 482.64±2.67 | 310.08±3.00 | 43.60±0.13 | 74.67±0.34 | 13.53±0.10 |
| Trembr | 505.62±4.57 | 376.88±7.34 | 48.99±0.38 | 72.08±0.29 | 17.01±0.50 | 473.97±1.24 | 301.45±4.98 | 44.50±0.35 | 75.11±0.67 | 12.90±0.74 |
| CTLE | 430.19±52.65 | 382.82±52.88 | 51.00±0.68 | 79.43±0.64 | 21.47±0.70 | 477.70±48.25 | 384.08±53.18 | 44.84±0.72 | 76.78±0.61 | 14.83±0.41 |
| Toast | 480.52±82.39 | 412.58±72.32 | 50.90±0.50 | 79.66±0.50 | 21.07±0.38 | 523.76±67.04 | 443.99±60.41 | 45.08±0.52 | 77.65±0.12 | 15.46±0.55 |
| TrajCL | 365.50±19.14 | 272.63±25.32 | 50.85±0.25 | 79.69±0.58 | 21.57±0.32 | 383.39±7.30 | 262.20±10.68 | 45.81±0.47 | 79.06±0.60 | 16.84±0.88 |
| LightPath | 553.27±42.26 | 360.86±56.41 | 49.15±0.23 | 78.59±0.58 | 20.66±0.27 | 598.20±15.57 | 348.61±19.32 | 44.39±0.25 | 72.75±0.47 | 14.42±0.54 |
| START | 333.10±10.47 | 240.40±15.10 | 52.78±0.31 | 80.42±0.41 | 23.32±0.31 | 319.00±4.27 | 208.35±7.30 | 46.13±0.27 | 79.34±0.49 | 16.31±1.36 |
| MMTEC | 312.78±44.13 | 212.78±39.26 | 52.85±0.29 | 79.55±0.39 | 22.69±0.40 | 311.99±3.44 | 207.94±3.65 | 47.45±0.14 | 79.45±0.26 | 17.90±0.24 |
| JGRM | 215.99±29.00 | 172.79±24.91 | 54.27±0.77 | 85.06±0.55 | 25.68±0.59 | 303.12±6.68 | 226.82±5.98 | 48.21±0.21 | 81.79±0.27 | 19.01±0.22 |
| **TrajMmaba** | **129.47±1.45** | **85.95±2.81** | **58.21±0.11** | **86.81±0.27** | **30.45±0.42** | **236.33±2.10** | **155.63±2.67** | **48.81±0.23** | **83.37±0.08** | **20.55±0.41** |

Table 10: Arrival Time Estimation (ATE) performance results with standard deviations. A lower value indicates better performance. **Bold**: the best, underline: the second best.

| Strategy | fine-tune / without fine-tune | | | | | |
| --- | --- | --- | --- | --- | --- | --- |
| Dataset | Chengdu | | | Xian | | |
| Method \ Metric | RMSE (seconds) | MAE (seconds) | MAPE (%) | RMSE (seconds) | MAE (seconds) | MAPE (%) |
| t2vec | 127.41±2.68 / 138.30±1.63 | 64.67±3.58 / 79.74±1.98 | 14.01±0.71 / 18.71±0.57 | 214.40±2.05 / 207.11±4.12 | 108.80±2.01 / 117.86±4.74 | 16.96±0.94 / 16.01±0.52 |
| Trembr | 124.32±3.67 / 159.60±8.40 | 63.42±0.57 / 110.36±6.51 | 13.60±0.28 / 29.50±1.00 | 209.12±3.02 / 435.04±5.17 | 107.02±1.39 / 337.35±3.66 | 16.40±0.86 / 47.11±0.06 |
| CTLE | 135.21±14.97 / 135.59±4.68 | 55.41±7.17 / 63.45±5.08 | 11.18±1.52 / 13.99±1.27 | 207.16±7.44 / 272.88±60.42 | 107.46±9.04 / 176.16±72.08 | 16.25±2.94 / 31.84±10.46 |
| Toast | 171.58±49.56 / 149.67±8.22 | 91.66±57.29 / 79.69±9.59 | 18.84±13.04 / 17.89±0.82 | 202.99±36.20 / 299.94±51.68 | 102.73±26.14 / 205.49±54.15 | 15.75±2.24 / 32.55±5.71 |
| TrajCL | 132.98±1.06 / 136.56±3.90 | 55.78±0.89 / 79.59±2.57 | 11.86±0.23 / 19.85±0.44 | 183.74±2.54 / 194.64±1.86 | 73.21±3.45 / 106.66±3.87 | 12.55±0.45 / 16.80±0.50 |
| LightPath | 123.00±8.85 / 129.48±0.26 | 58.04±8.56 / 56.82±2.58 | 12.83±1.70 / 12.71±1.00 | 169.01±1.94 / 186.02±3.96 | 74.08±3.13 / 77.33±2.59 | 10.50±0.41 / 10.41±0.38 |
| START | 121.11±16.25 / 144.54±0.90 | 58.97±11.59 / 79.78±0.87 | 13.49±2.57 / 19.72±0.27 | 159.89±4.55 / 213.22±2.19 | 72.19±3.09 / 120.74±2.70 | 10.26±0.35 / 20.01±0.49 |
| MMTEC | 105.72±4.74 / 124.98±2.50 | 40.48±1.46 / 54.76±1.64 | 8.17±0.51 / 12.13±0.69 | 155.59±17.66 / 183.66±1.44 | 58.84±6.54 / 78.14±4.33 | 7.65±0.98 / 12.03±0.87 |
| JGRM | 65.24±8.45 / 119.40±1.08 | 31.52±2.96 / 82.02±0.99 | 6.84±0.69 / 22.01±0.70 | 84.49±3.88 / 202.81±2.04 | 39.92±1.55 / 137.75±0.40 | 5.97±0.48 / 20.20±0.43 |
| **TrajMmaba** | **50.17±0.67 / 92.52±4.27** | **17.73±0.29 / 44.89±2.62** | **4.77±0.13 / 11.07±0.73** | **74.75±0.95 / 134.57±5.52** | **25.99±2.56 / 55.74±1.67** | **4.55±0.90 / 8.79±0.33** |

Table 11: Similar Trajectory Search (STS) performance results with standard deviations. ↑: higher better, ↓: lower better. **Bold**: the best, underline: the second best.

| Dataset | Chengdu / Xian | | |
| --- | --- | --- | --- |
| Method \ Metric | Acc@1 ↑ (%) | Acc@5 ↑ (%) | Mean Rank ↓ |
| t2vec | 75.20±1.51 / 89.83±0.55 | 88.53±0.76 / 94.87±0.55 | 5.04±0.47 / 5.71±0.33 |
| Trembr | 84.60±1.06 / 78.75±6.40 | 89.63±0.40 / 86.45±4.07 | 10.01±2.27 / 11.04±3.03 |
| CTLE | 69.07±4.24 / 46.80±6.93 | 78.50±1.91 / 58.15±5.02 | 41.69±5.19 / 48.32±7.40 |
| Toast | 71.73±3.88 / 32.60±2.40 | 81.70±3.18 / 60.00±8.91 | 46.16±2.11 / 46.02±6.81 |
| TrajCL | 94.45±0.07 / 90.87±0.06 | 96.70±0.14 / 94.50±0.10 | 2.88±0.36 / 4.86±1.08 |
| LightPath | 68.35±5.73 / 74.50±2.97 | 82.05±6.86 / 88.75±1.91 | 17.94±9.92 / 6.99±0.40 |
| START | 91.07±0.15 / 91.43±0.21 | 96.57±0.21 / 96.23±0.50 | 4.90±1.39 / 4.26±1.15 |
| MMTEC | 90.17±0.12 / 82.97±0.06 | 96.47±0.06 / 91.17±0.06 | 3.76±1.48 / 4.86±0.35 |
| JGRM | 86.63±1.01 / 86.97±1.45 | 96.10±0.44 / 96.77±0.76 | 1.97±0.13 / 1.62±0.12 |
| **TrajMmaba** | **96.53±0.38 / 97.97±0.12** | **99.13±0.17 / 99.80±0.00** | **1.15±0.04 / 1.07±0.03** |

# F Computation Cost

We analyze the time complexity of TrajMamba's downstream components, i.e., mask generator and Traj-Mamba encoder, and compare them with typical baselines JGRM and START.

**Inference Time Complexity of TrajMamba.** For the mask generator, the time complexity is $O(n'd^2)$, where $d$ is the number of trajectory point features for the input mask generator and $n'$ is the length of the input trajectory after preprocessing. For the Traj-Mamba Encoder, the time complexity is $O(\tilde{n}dE)$ and $O(L\tilde{n}E^2)$ for embedding initial latent vector and $L$ stacked Traj-Mamba blocks, respectively. Here, $\tilde{n}$ is the length of the compressed trajectory output by the mask generator and $E$ is the embedding dimension. Therefore, the combined time complexity of TrajMamba is $O(n'd^2 + \tilde{n}dE + L\tilde{n}E^2)$, where $n', \tilde{n} \propto n$ and $n$ is raw trajectory length.

**Complexity of Typical Baselines.** For baseline JGRM, the time complexity is $O(mE^2 + L_r m^2 E + L_f(2m)^2 E)$, where $m$ is the number of unique road segments in trajectories, and $L_r, L_f$ are the numbers of route encoder layers and model interaction layers, respectively. And for START, the time complexity is $O(L_g H|\mathcal{V}|E^2 + L_e n^2 E)$, where $|\mathcal{V}|$ is the number of road segments in the road network, $H$ is the number of attention heads used by TPE-GAT, and $L_g, L_e$ are the numbers of TPE-GAT layers and TAT-Enc layers, respectively.

It is clear that TrajMamba achieves linear inference time complexity with respect to trajectory length $n$ by avoiding the quadratic terms present in baselines: JGRM introduces $O(m^2)$ from Transformer-based modules, while START relies on $O(n^2)$. This design ensures encoding efficiency for long or large-batch trajectories in real-world scenarios, which is crucial for real-time applications.

# G Variants of Ablation

To assess the effectiveness of the modules implemented in TrajMamba, we compare the complete model with the following variants:

- *V-Mamba*: replace the Traj-Mamba blocks with the vanilla Mamba2 blocks.
- *V-Transformer*: replace all Mamba-based modules with the vanilla Transformer layers.
- *w/o Purpose*: replace the travel purpose-aware pre-training with a reconstruction task to pre-train the teacher Traj-Mamba encoder.
- *w/o KD*: remove the knowledge distillation pre-training and directly align compressed representations with the road and POI views derived from full trajectories.
- *w/o Compress*: remove the mask generator and directly use full-trajectory embeddings for downstream tasks.
- *MG-Trans-DP*: replace the mask generator with the Douglas-Peucker algorithm to generate compressed trajectories.
- *MG-Trans-DS*: replace the mask generator with direct trajectory downsampling at a 60% sampling ratio.
- *w/o Filter*: remove rule-based filtering preprocessing used in mask generator.

## H  Trajectory Spatio-temporal Feature Embedding

Given a trajectory $\mathcal{T} = \langle \tau_1, \tau_2, \ldots, \tau_n \rangle$, we embed its GPS and road perspectives into a $\frac{E}{2}$-dimensional embedding space respectively, where $E$ represents the embedding dimension of the trajectory embeddings.

For the GPS coordinates $g_i$ of each trajectory point $\tau_i$, we employ a linear transformation layer to map it to $\mathbb{R}^{\frac{E}{2}}$. The road segment $e_i$ is also mapped to $\mathbb{R}^{\frac{E}{2}}$ via an index-fetching embedding layer $\boldsymbol{E}_{\mathrm{idx}}$ and a linear projection. Then we transform the timestamp $t_i$ into five features: two duration time features $\boldsymbol{t}_i^{\mathrm{dur}}$ including the time delta $\Delta t_i$ in minutes relative to $t_1$ and the timestamp in minutes; three cyclic time features $\boldsymbol{t}_i^{\mathrm{cyc}}$ including day-in-week, hour-in-day, and minute-in-hour. These features are then encoded into five embedding vectors using learnable Fourier encoding layers [35]. Next, the two duration time feature vectors are concatenated and then mapped into $\mathbb{R}^{\frac{E}{2}}$ through a linear layer. The three cyclic time feature vectors also undergo the same process. Finally, the GPS latent vector $\boldsymbol{z}_i^G$ and road latent vector $\boldsymbol{z}_i^R$ of the point are obtained as follows:

$$\begin{aligned}
\boldsymbol{z}_i^G &= \mathrm{Linear}(g_i) + \mathrm{Linear}(\mathrm{Cat}(\mathrm{Fourier}(\boldsymbol{t}_i^{\mathrm{dur}}))) \\
\boldsymbol{z}_i^R &= \mathrm{Linear}(\boldsymbol{E}_{\mathrm{idx}}(e_i)) + \mathrm{Linear}(\mathrm{Cat}(\mathrm{Fourier}(\boldsymbol{t}_i^{\mathrm{cyc}}))),
\end{aligned} \tag{11}$$

where $\mathrm{Cat}$ denotes vector concatenation, and $\mathrm{Fourier}$ denotes the learnable Fourier encoding layer. By gathering the GPS and road latent vectors for each point in $\mathcal{T}$, respectively, we obtain its two sequences of latent vectors as $\boldsymbol{Z}_{\mathcal{T}}^G = \langle \boldsymbol{z}_1^G, \boldsymbol{z}_2^G, \ldots, \boldsymbol{z}_n^G \rangle \in \mathbb{R}^{n \times \frac{E}{2}}$ and $\boldsymbol{Z}_{\mathcal{T}}^R = \langle \boldsymbol{z}_1^R, \boldsymbol{z}_2^R, \ldots, \boldsymbol{z}_n^R \rangle \in \mathbb{R}^{n \times \frac{E}{2}}$.

We also calculate the high-order movement behavior features for each point, including speed $v_i$, acceleration $\mathrm{acc}_i$, and movement angle $\theta_i$, according to the difference between the features of $\tau_i$ and $\tau_{i+1}$. Note that these features of the last point $\tau_n$ are set to 0. Then these three features are min-max normalized and concatenated into a vector denoted as $\boldsymbol{s}_i = (v_i, \mathrm{acc}_i, \theta_i)$. By gathering this vector for each point in $\mathcal{T}$, we obtain its sequence of high-order movement behavior features as $\boldsymbol{S}_{\mathcal{T}} = \langle \boldsymbol{s}_1, \boldsymbol{s}_2, \ldots, \boldsymbol{s}_n \rangle \in \mathbb{R}^{n \times 3}$.

## I  Road/POI Textual Embedding

Given a trajectory $\mathcal{T} = \langle \tau_1, \tau_2, \ldots, \tau_n \rangle$ and the road network $\mathcal{G}$, the initial textual embeddings $\boldsymbol{z}_{e_i}, \boldsymbol{z}_{p_i} \in \mathbb{R}^E$ for each road segment $e_i$ and POI $p_i$ corresponding to point $\tau_i$ are obtained as follows:

$$\boldsymbol{z}_{e_i} = \mathrm{Linear}(\boldsymbol{E}_{\mathrm{text}}(\mathrm{desc}_i^{\mathrm{Road}})), \ \boldsymbol{z}_{p_i} = \mathrm{Linear}(\boldsymbol{E}_{\mathrm{text}}(\mathrm{desc}_i^{\mathrm{POI}})), \tag{12}$$

where $\boldsymbol{E}_{\mathrm{text}}$ denotes a shared pre-trained textual embedding module for mapping a line of text into an embedding vector, for which we use the *text-embedding-3-large* model provided by OpenAI[6].

Next, we aggregate the global information from the origin $\tau_1$ and destination $\tau_n$ as well as the local information from neighboring road segments and POIs to update $\boldsymbol{z}_{e_i}$ and $\boldsymbol{z}_{p_i}$ for futher semantic

---

[6]https://platform.openai.com/docs/guides/embeddings

capture. Specifically, we obtain the neighbor set $\mathcal{N}_i$ of road segment $e_i$ according to $\mathcal{G}$; while for the neighbor set $\mathcal{N}'_i$ of POI $p_i$, we consider POIs within a radius of $R$ from $p_i$ and select the closest 10 if the number exceeds. And we implement aggregation functions $\mathrm{Agg}^{\mathrm{Road}}(\cdot), \mathrm{Agg}^{\mathrm{POI}}(\cdot)$ in Eq. 3 as follows:

$$
\begin{aligned}
\mathrm{Agg}^{\mathrm{Road}}(\boldsymbol{z}_{e_j}, \boldsymbol{z}_{e_1}, \boldsymbol{z}_{e_n}|j \in \mathcal{N}_i) &= \sigma_r(\mathrm{BN}(\mathbf{W}_1 \sum\nolimits_{j \in \mathcal{N}_i} w_j \boldsymbol{z}_{e_j} + \mathbf{W}_2(w_i^o \boldsymbol{z}_{e_1} + w_i^d \boldsymbol{z}_{e_n}))) \\
\mathrm{Agg}^{\mathrm{POI}}(\boldsymbol{z}_{p_j}, \boldsymbol{z}_{p_1}, \boldsymbol{z}_{p_n}|j \in \mathcal{N}'_i) &= \sigma_r(\mathrm{BN}(\mathbf{W}_3 \sum\nolimits_{j \in \mathcal{N}'_i} w'_j \boldsymbol{z}_{p_j} + \mathbf{W}_4(w_i^o \boldsymbol{z}_{p_1} + w_i^d \boldsymbol{z}_{p_n}))),
\end{aligned}
\tag{13}
$$

where $\mathbf{W}_1, \ldots, \mathbf{W}_4 \in \mathbb{R}^{E \times E}$ are learnable parameters, $\sigma_r$ denotes ReLU activation function, and BN represents batch normalization. The contribution weights $w_i^o, w_i^d$ of $\tau_1$ and $\tau_n$ depend on the time interval between $\tau_i$ and each of them, i.e., $w_i^d = \Delta t_i / \Delta t_n, w_i^o = 1 - w_d$. The weight $w_j$ of the neighbor road segment $e_j$ depends on its textual relevance to $e_i$, calculated by a global linear attention mechanism $\mathrm{Att}(\cdot, \cdot)$, and the transition probability $\varphi_{ij}$ from $e_i$ to $e_j$ derived from all observed trajectories. Similarly, the weight $w'_j$ of the neighbor POI $p_j$ is affected by its textual relevance to $p_i$ and their distance $dist_{ij}$. Formally:

$$
\begin{aligned}
w_j &= \mathrm{Norm}_{L1}(\mathrm{Att}(\boldsymbol{z}_{e_i}, \boldsymbol{z}_{e_j})) + \alpha \varphi_{ij} \\
w'_j &= \mathrm{Norm}_{L1}(\mathrm{Att}(\boldsymbol{z}_{p_i}, \boldsymbol{z}_{p_j})) + \beta \mathrm{Norm}_{L1}(f(dist_{ij})) \\
\mathrm{Att}(\boldsymbol{z}_i, \boldsymbol{z}_j) &= \exp(\mathbf{v}^\top \tanh(\mathrm{Linear}(\mathrm{Cat}(\boldsymbol{z}_i, \boldsymbol{z}_j)))) \\
f(dist_{ij}) &= \exp(-dist_{ij} / \max_{k \in \mathcal{N}_i}(dist_{ik})),
\end{aligned}
\tag{14}
$$

where $f(dist_{ij})$ is the distance weighting function, $\mathrm{Norm}_{L1}(\cdot)$ is the L1 normalization based on the corresponding neighbor set, $\mathbf{v} \in \mathbb{R}^E$ is a learnable parameter, and $\alpha, \beta$ are hyper-parameters.

Futhermore, to explore the impact of pre-trained textual embedding module on our model's performance, we introduce two additional pre-trained textual embedding modules provided by HuggingFace[7], i.e., *hfl/chinese-lert-large* and *BAAI/bge-large-zh-noinstruct*. As shown in Tab. 12, using alternative models has limited impact on performance, demonstrating that our pre-training process enables the model to learn robust representations.

Table 12: Impact of pre-trained textual embedding module $\boldsymbol{E}_{\text{text}}$ on Chengdu dataset in all tasks. $\uparrow$: higher better, $\downarrow$: lower better. **Bold**: the best, underline: the second best.

| Task | | Destination Prediction | | | | Arrival Time Estimation | | | Similar Trajectory Search | | |
|---|---|---|---|---|---|---|---|---|---|---|---|
| Metric | GPS | | Road Segment | | | RMSE ↓ | MAE ↓ | MAPE ↓ | Acc@1 ↑ | Acc@5 ↑ | Mean ↓ |
| Method | RMSE ↓ (meters) | MAE ↓ (meters) | Acc@1 ↑ (%) | Acc@5 ↑ (%) | Recall ↑ (%) | (seconds) | (seconds) | (%) | (%) | (%) | Rank |
| chinese-lert-large | **127.97** | **83.73** | 58.22 | 86.72 | **30.58** | 50.25 | 17.76 | 4.65 | **97.30** | 99.30 | **1.13** |
| bge-large-zh-noinstruct | 131.34 | 87.63 | **58.39** | 86.80 | 30.30 | 50.32 | **17.46** | **4.63** | 96.30 | **99.40** | 1.15 |
| **text-embedding-3-large** | 129.47 | 85.95 | 58.21 | **86.81** | 30.45 | **50.17** | 17.73 | 4.77 | 96.53 | 99.13 | 1.15 |

---

[7]https://huggingface.co

