# OpenReview forum: "TrajMamba: An Efficient and Semantic-rich Vehicle Trajectory Pre-training Model"
_NeurIPS.cc/2025/Conference — NeurIPS 2025 spotlight_

### Official Review · Reviewer_ewCK · 2025-06-30

**Clarity:** 2
**Significance:** 3
**Originality:** 2
**Rating:** 4
**Confidence:** 5

**Summary:**

This paper proposes an efficient and semantically rich pre-trained model for vehicle trajectory design, TrajMamba. The model primarily addresses two core challenges: First, efficiently extracting travel purposes related to road and POI features; Second, how to handle redundant data points in real trajectories to improve computational efficiency and embedding quality. Specifically, TrajMamba comprises three key components:  Traj-Mamba Encoder jointly models GPS and road perspective to capture complex vehicle movement patterns.  Travel Purpose-Aware Pre-training aligns the trajectory embeddings learned by the model with travel purposes extracted from road and POI text information, enabling the model to understand travel intent without incurring additional computational overhead. Knowledge Distillation Pre-training uses a learnable masking generator to identify and compress key points in trajectories, yielding efficient and information-rich compressed trajectory embeddings.   Extensive experiments on two real-world datasets and three downstream tasks demonstrate that TrajMamba outperforms current state-of-the-art baseline models in both accuracy and efficiency.

**Questions:**

1. Regarding the trajectory compressor or mask generator, does it mainly remove redundant trajectory points? What is the effect of directly downsampling the trajectory by a certain ratio?
2. Regarding the task of arrival time estimation, the setting seems somewhat unreasonable. Typically, trajectory points have similar sampling frequencies, and predictions based on start and end points or road segment sequences should be more realistic.

**Ethical Concerns:**

["NO or VERY MINOR ethics concerns only"]

**Final Justification:**

The overall quality of this work is satisfactory, but the generalization performance and efficiency issues of the model still further limit its applicability.

**Limitations:**

yes

**Quality:**

3

**Strengths And Weaknesses:**

Strengths:
1. TrajMamba can effectively extract two key types of semantic information from vehicle trajectories. The first is to capture continuous “movement patterns” by jointly modeling GPS and road perspectives. The second is to efficiently learn “travel purposes” from road and POI information through pre-training.
2. The proposed model uses knowledge distillation pre-training and a learnable mask generator to effectively identify and compress redundant points in trajectories, thereby improving computational efficiency and embedding quality.
3. The experiments conducted on two real-world datasets and three downstream tasks demonstrated that TrajMamba outperforms existing state-of-the-art baseline models and exhibits efficiency in terms of model size and computation time.
4. The proposed model offers scalability. Compared with other advanced models, TrajMamba can achieve better performance with less training data and shorter fine-tuning time.


Weaknesses：
1. The performance of the model depends heavily on high-quality external data, including accurate road network maps, POI information, and the accuracy of the map-matching algorithm. If these external data contain noise or are missing (which is common for trajectory data), the performance of the model may decline.
2. This paper proposes a two-stage pre-training process that includes “purpose perception” and “knowledge distillation.” However, the article does not fully argue why these two independent stages are necessary, and whether training directly using compressed trajectories can achieve the same effect.
3. This paper claims that the model is efficient, mainly in terms of the final embedding computation (inference) time and model size. However, this paper uses a two-stage training framework and combines many modules and additional information collection operations, the efficiency of which may not be satisfactory and may mislead readers' judgment.
4. The experiment was conducted only on trajectories in two Chinese cities (Chengdu and Xi'an) and provided by the same organization. The homogeneity of this dataset raises questions about the model's generalization ability.

---

> ### Author Rebuttal · Authors · 2025-07-30
>
> We sincerely appreciate your detailed and insightful comments, which are highly valuable for improving our work. We have carefully addressed each point below.
>
> **[W1]** Here is our clarification of external data quality's influence on performance, based on the actual design and implementation:
>
> - Our model does rely on city road networks and POI data, which are independent of trajectory data. However, these can be easily obtained through the following means: (1) Road networks from OpenStreetMap, a widely used open-source platform with stable data quality. (2) POI information from reliable map APIs like AMap and Google Maps. Such sources are widely available and effective, ensuring stable and high-quality data that is less prone to noise or missing information. For our experimental datasets, we directly obtained these external data through the above channels.
>
> - Regarding the accuracy of map-matching algorithms, it is common practice to use such algorithms (e.g., FMM[1], LMM[2]) to map GPS trajectories to urban road networks. Most recent map-matching algorithms have strong capabilities to handle noisy data and complex road networks.
>
>   [1] Can Yang and Gyozo Gidofalvi. 2018. Fast map matching, an algorithm integrating hidden Markov model with precomputation. International Journal of Geographical Information Science 32, 3 (2018), 547–570.
>
>   [2] Wannes Meert and Mathias Verbeke. 2018. HMM with non-emitting states for Map Matching. In ECDA.
>
> **[W2]** This design is rooted in the **teacher-student framework** of knowledge distillation, where a well-trained teacher model transfers knowledge to a student model. Specifically:
>
> - The travel purpose-aware pre-trained teacher model first learns semantically-rich embeddings from **complete trajectories**, capturing both movement patterns and travel purposes. This aligns with the principle that a knowledgeable teacher is essential for effective knowledge transfer.
> - In knowledge distillation pre-training stage, the student model then learns to generate compressed trajectory representations guided by the teacher's output embeddings. This avoids the instability of training two uninformed student models to learn from each other. They may fall into the local optimal solution during training, unable to obtain an effective learning direction, which will lead to training collapse.
>
> Moreover, we designed an ablation variant **"w/o KD"** in our ablation study, which removes the knowledge distillation pre-training and **directly trains compressed representations through travel purpose-aware pre-training**. Its worse performance compared to our complete model demonstrates that the two-stage pre-training procedure is necessary and effective for TrajMamba's performance.
>
> **[W3]** Our efficiency focus is explicitly on **downstream inference**—the phase critical for real-world applications (e.g., real-time trajectory encoding), rather than pre-training.
>
> The two-stage training, along with additional modules and information collection, are part of the offline pre-training process. Pre-training involves more flexible resource allocation, so their computational costs are acceptable. Importantly, these components do not participate in downstream tasks: they are only used during model training, leaving no additional overhead in real-world applications.
>
> This design ensures that the efficiency metrics we highlight—final embedding computation time—accurately reflect the performance users will experience in deployment, avoiding the persistent downstream costs that would arise from alternative designs.
>
> **[W4]** To address this, we conducted supplementary experiments on a new real-world trajectory dataset: **Porto**. Hosted on Kaggle for a taxi trajectory prediction contest, this dataset is widely used in TRL studies and contains GPS trajectories of taxis in Porto, Portugal. We obtained POI information from the Azure Maps API and road networks from OpenStreetMap.
>
> We re-evaluated our model and several best-performing baselines from the Chengdu and Xi'an datasets, including **START, MMTEC, and JGRM**, on the Porto dataset across all downstream tasks. The results demonstrate that our model maintains consistent performance advantages over baselines on the Porto dataset, aligning with its behavior on the Chengdu and Xi'an datasets. These experimental results verify our model's generalization ability across trajectories from different geographical regions and data sources.
>
> | Task      |             | Destination | Prediction  |             |            | Arrival    | Time      | Estimation | Similar   | Trajectory  | Search      |
> | --------- | ----------- | ----------- | ----------- | ----------- | ---------- | ---------- | --------- | ---------- | --------- | ----------- | ----------- |
> | Metric    | RMSE        | MAE         | Acc@1       | Acc@5       | Recall     | RMSE       | MAE       | MAPE       | Mean Rank | ACC@1       | ACC@5       |
> | START     | 329.393     | 222.210     | 23.655%     | 51.459%     | 6.692%     | 15.435     | 5.134     | 0.860%     | 7.398     | 83.500%     | 88.800%     |
> | MMTEC     | 331.370     | 225.183     | 23.569%     | 51.065%     | 6.369%     | 17.193     | 7.413     | 1.124%     | 5.160     | 81.900%     | 89.600%     |
> | JGRM      | 315.479     | 217.562     | 23.981%     | 51.643%     | 6.804%     | 13.584     | 5.384     | 0.864%     | 2.156     | 88.900%     | 95.800%     |
> | TrajMamba | **296.413** | **194.835** | **24.156%** | **52.464%** | **7.696%** | **10.680** | **4.417** | **0.768%** | **1.664** | **94.800%** | **99.300%** |
>
> **[Q1]** Yes, our mask generator primarily identifies key trajectory points and removes redundant ones. We conducted supplementary experiments to compare its effect with directly downsampling trajectories by a certain ratio (sampling ratio=60%) across all downstream tasks on the Chengdu dataset. These results verify our mask generator's ability to identify key trajectory points for effective compression, which is superior to direct downsampling.
>
> | Task                  |             | Destination | Prediction  |             |             | Arrival    | Time       | Estimation | Similar   | Trajectory  | Search      |
> | --------------------- | ----------- | ----------- | ----------- | ----------- | ----------- | ---------- | ---------- | ---------- | --------- | ----------- | ----------- |
> | Metric                | RMSE        | MAE         | Acc@1       | Acc@5       | Recall      | RMSE       | MAE        | MAPE       | Mean Rank | ACC@1       | ACC@5       |
> | directly downsampling | 150.700     | 106.353     | 53.124%     | 80.460%     | 23.158%     | 54.695     | 19.553     | 5.562%     | 2.161     | 95.100%     | 98.800%     |
> | TrajMamba             | **129.474** | **85.947**  | **58.205%** | **86.813%** | **30.446%** | **50.166** | **17.727** | **4.772%** | **1.148** | **96.525%** | **99.125%** |
>
> Additionally, we designed an ablation variant **"w/o MG"** in our ablation study, which replaces the mask generator with the Douglas-Peucker algorithm to generate compressed trajectories. Its worse performance compared to our complete model in the DP task also indicates the advantages of our mask generator for various downstream tasks.
>
> **[Q2]** In our experimental datasets (Chengdu and Xi'an), trajectory points do not have strictly uniform sampling intervals—unlike the Porto dataset, which has a fixed 15-second interval. While most trajectories have sampling intervals of at least 6 seconds, there is significant variability:
>
> | Dataset                                          | Chengdu | Xian   |
> | ------------------------------------------------ | ------- | ------ |
> | Average sampling interval                        | 6.33    | 12.55  |
> | Median sampling interval                         | 6.00    | 12.00  |
> | Standard deviation of sampling interval          | 13.49   | 22.31  |
> | Proportion of sampling interval <6s / <12s       | 5.59%   | 7.48%  |
> | Proportion of sampling interval 6s / 12s         | 88.27%  | 83.56% |
> | Proportion of sampling interval 7\~10s / 13\~15s | 4.85%   | 6.92%  |
> | Proportion of sampling interval >10s / >15s      | 1.27%   | 2.03%  |
>
> Moreover, our ATE task is designed to predict the end time of **trajectories still in movement**. This setting aligns with practical scenarios such as ride-hailing arrival time estimation and traffic management, where real-time estimation of remaining travel time for ongoing trips holds substantial application value. These variations and task design choices reflect real-world trajectory data characteristics and practical needs, enhancing the model's adaptability to real-world scenarios.
>
> Thank you again for your earnest efforts in identifying weaknesses and raising relevant questions. We hope these supplements meet your expectations and look forward to further suggestions for improvement.

---

> > ### Comment · Reviewer_ewCK · 2025-08-04
> >
> > I appreciate the reviewers' hard work and response, and I no longer have any concerns.

---

> > > ### Author Response · Authors · 2025-08-04
> > >
> > > We're glad that our response addressed your concerns. We truly appreciate your time and feedback.

---

### Official Review · Reviewer_9qTq · 2025-06-30

**Clarity:** 3
**Significance:** 3
**Originality:** 3
**Rating:** 6
**Confidence:** 4

**Summary:**

This paper proposes an efficient and semantics-rich vehicle-trajectory learning called TrajMamba which consists of three coordinated components. On two DiDi taxi datasets (Chengdu & Xi’an), TrajMamba improves over nine SOTA baselines on three downstream tasks—destination prediction, arrival-time estimation, and similar-trajectory search.

**Questions:**

(1) Do the other methods in Table 7 also have pre-training stage? If they do not have a separate pre-training stage, what is the setting and is this fair?

(2) See weakness.

**Ethical Concerns:**

["NO or VERY MINOR ethics concerns only"]

**Final Justification:**

The authors’ response fully addresses all issues, so I will raise my rating to strong accept.

**Limitations:**

Yes.

**Paper Formatting Concerns:**

None.

**Quality:**

3

**Strengths And Weaknesses:**

Strengths

(1) The work tackles the real-world bottleneck of scaling semantic trajectory analytics, efficiently modelling GPS traces while preserving travel-purpose information.

(2) By replacing quadratic self-attention with linear SSM blocks and adding a learnable compression module, the whole pipeline keeps time- and memory-costs low.

(3) Three downstream tasks (destination prediction, arrival-time estimation, similar-trajectory search) are evaluated on two large-scale city datasets, with comprehensive ablations, sensitivity studies and efficiency baselines.

Weaknesses

(1) DP task only computes the MAE/RMSE for the last coordinate and path-aware metrics are absent, which may not reflect the exact learning ability of the trajectory representation.

(2) In Table 7, the “w/o FT” variant underperforms fine-tuned models by a large margin. Without stronger evidence, it is difficult to assess the intrinsic value of the pre-training stage.

---

> ### Author Rebuttal · Authors · 2025-07-29
>
> We sincerely appreciate your recognition of our work—this means a great deal to us. Below, we address your doubts as follows.
>
> **[W1]** For the DP task, we believe our paper already employs path-aware metrics for performance evaluation. In Section 5.1, beyond computing the MAE/RMSE for the last coordinate, we also predict trajectory destinations **at the road segment level** to assess model performance. This road segment prediction is treated as a multi-classification task, optimized via cross-entropy loss, with evaluation metrics including Accuracy@N (Acc@1, Acc@5) and macro-Recall. We consider these metrics as 'path-aware metrics' because road segments are intrinsic components of the path itself. Predicting road segments and evaluating via these indicators directly reflects our model's ability to learn and predict path-related information. Compared to relying solely on MAE/RMSE for final coordinates, this approach enables a more comprehensive evaluation of the model's capability in the DP task across multiple dimensions. Detailed results are presented in Tables 1, 7, and 8. If our understanding of "path-aware metrics" differs from your expectation, we sincerely apologize and look forward to your clarification in the second round of review.
>
> **[W2/Q1]** For the without fine-tune setting used in our experiments, we clarify that all models under this setting fix their parameters after the pre-training stage and only update the predictors' parameters. In other words, the other methods in Table 7 also have a pre-training stage, and all models simply do not participate in fine-tuning with task supervision (only predictors are trained). To assess the intrinsic value of the pre-training stage, the variants **"w/o Purpose"** and **"w/o KD"** in our ablation study remove the two training stages proposed in the paper (for DP and ATE tasks under the fine-tune setting), and their worse performance compared to our complete model demonstrates that both pre-training procedures contribute to TrajMamba's performance. Moreover, we supplement the performance of these **variants under the w/o ft setting on the Chengdu dataset** to further demonstrate the effectiveness of the two training stages.
>
> | Task        |             | Destination | Prediction  |             |             | Arrival    | Time       | Estimation  |
> | ----------- | ----------- | ----------- | ----------- | ----------- | ----------- | ---------- | ---------- | ----------- |
> | Metric      | RMSE        | MAE         | Acc@1       | Acc@5       | Recall      | RMSE       | MAE        | MAPE        |
> | w/o Purpose | 405.480     | 330.212     | 51.392%     | 77.038%     | 22.023%     | 106.729    | 52.578     | 12.931%     |
> | w/o KD      | 639.438     | 512.874     | 41.778%     | 68.915%     | 14.376%     | 118.398    | 65.387     | 15.548%     |
> | TrajMamba   | **365.602** | **291.119** | **51.954%** | **77.722%** | **22.953%** | **92.519** | **44.889** | **11.073%** |
>
> We sincerely hope that the above content has adequately addressed your questions and concerns. Your positive feedback has been highly encouraging, and we look forward to further enhancing the work with your continued guidance.

---

> > ### Comment · Reviewer_9qTq · 2025-08-04
> >
> > The authors’ response fully addresses all issues, so I will raise my rating to strong accept.

---

> > > ### Author Response · Authors · 2025-08-04
> > >
> > > Thank you again for your recognition of our work, and if you have any other questions, please feel free to contact us.

---

### Official Review · Reviewer_e8aq · 2025-07-01

**Clarity:** 3
**Significance:** 2
**Originality:** 2
**Rating:** 4
**Confidence:** 3

**Summary:**

This paper studies the problem of vehicle trajectory learning. To address it, this paper proposes TrajMamba, a semantic-rich trajectory learning framework that integrates GPS and road features via a dual-branch encoder, leverages travel purpose and applies knowledge distillation for trajectory compression. Experiments on two datasets demonstrate the effectiveness of proposed model.

**Questions:**

Q1: In the Hyperparameter Analysis section, shouldn’t the optimal value of E be 512 instead of 256 based on the results?

Q2: For the DP task on road segments, why is Recall used as the evaluation metric instead of F1-score, which is more commonly adopted in the literature [4]?

**Ethical Concerns:**

["NO or VERY MINOR ethics concerns only"]

**Final Justification:**

The authors have addressed my comments in general. I have raised my score accordingly.

**Limitations:**

Yes

**Quality:**

2

**Strengths And Weaknesses:**

Strengths:
S1: This paper investigates the fundamental vehicle trajectory learning problem.
S2: This paper develops an efficient and semantic-rich vehicle trajectory pretraining model.
S3: Experiments are conducted to demonstrate the effectiveness of the proposed method.

Weaknesses:

W1: The motivation of this paper is weak.
(1) The claimed burden of textual modeling for extracting travel purposes seems overstated. Lightweight or offline text encoders (e.g., ALBERT-12M, MiniLM-22M, DistilBERT-66M) are widely available and effective, yet the paper neither adopts nor compares against these alternatives. Besides, the process for obtaining the initial textual embeddings in the Travel Purpose-aware Pre-training module is not clear. If these embeddings are derived using the above lightweight or offline encoders, the motivation that "extracting travel purposes can be computationally heavy" is questionable. Moreover, although the authors claim no additional overhead, the proposed dual-text encoder setup inherently introduces latent computational costs, making the efficiency claims difficult to verify.
(2) The authors argue that redundant GPS points harm both efficiency and effectiveness, yet the claim regarding effectiveness is not well substantiated.

W2: Although the paper presents some technical contributions, the proposed Traj-Mamba framework exhibits limited novelty, as it primarily adopts existing Mamba/SSM blocks—which have already been explored in trajectory modeling [1]—to model GPS and road views separately, followed by a simple linear fusion. Besides, this dual-view design for GPS and road views  is also a common paradigm in trajectory representation learning.

[1] "Trajectory mamba: Efficient attention-mamba forecasting model based on selective ssm." Proceedings of the Computer Vision and Pattern Recognition Conference. 2025.

W3: The claimed effectiveness of the compression module is weakened by the use of existing rule-based preprocessing to remove explicit redundancy. This setup makes it difficult to disentangle the contribution of the proposed soft masking mechanism from that of the preprocessing itself. I recommend providing an case study to better validate the effectiveness of the learnable trajectory compression.

W4. To justify the paper’s claim of achieving state-of-the-art performance, it is recommended to include recent work such as [2, 3,4] as baselines.

[2] "Mrgrp: Empowering courier route prediction in food delivery service with multi-relational graph." Companion Proceedings of the ACM on Web Conference. 2025.
[3] "UniTR: A Unified Framework for Joint Representation Learning of Trajectories and Road Networks." Proceedings of the AAAI Conference on Artificial Intelligence. 2025.
[4] "Path-LLM: A Multi-Modal Path Representation Learning by Aligning and Fusing with Large Language Models." Proceedings of the ACM on Web Conference. 2025.

W5. The paper emphasizes efficiency, but the observed improvements are modest. Besides, a detailed complexity analysis is recommended to support the efficiency claims.

---

> ### Author Rebuttal · Authors · 2025-07-31
>
> We sincerely appreciate your detailed and insightful comments, which are highly valuable for improving our work. We have carefully addressed each point below.
>
> **[W1]** We clarify the motivation as follows:
>
> (1) Our focus is **eliminating additional downstream overhead**, which is more critical for real-world applications (e.g., real-time trajectory encoding), rather than pre-training costs. As noted, pre-training is offline with flexible resources, so its computational costs are acceptable if enabling effective downstream inference. Addressing your concerns:
>
> - The initial textual embeddings are obtained using OpenAI's text-embedding-3-large model (See Appendix H). We apologize for omitting this.
> - Regarding lightweight text encoders, we acknowledge their effectiveness but emphasize they conflict with our downstream efficiency goal: (i) Their sizes (e.g., 45.8, 83.8, and 251.7MB) are 8–47× larger than our downstream module (5.407/5.182MB), increasing deployment memory. (ii) They typically require 1024+ hidden dimensions to encode complex textual information, while our model (and most standard trajectory learning models) uses approximately 256. Integrating them into downstream tasks would inevitably add non-negligible overhead.
> - The dual-text encoder is exclusive to offline pre-training and does not participate in downstream tasks. Therefore, it introduces no additional overhead in downstream inference, avoiding persistent computational costs.
>
> (2) The variant **"w/o Compress"** without compression in our ablation study shows consistent performance degradation across all downstream tasks, directly verifying that redundant GPS points harm effectiveness.
>
> **[W2]** We appreciate your attention to the novelty of the Traj-Mamba Encoder. The referenced Trajectory Mamba[1] and our TrajMamba differ fundamentally in their core goals and technical designs, which we clarify as follows:
>
> 1. Trajectory Mamba is tailored for real-time autonomous driving decision-making, where the focus lies in short-term (seconds) behavior prediction at intersections. Its data is often high-frequency (10-20Hz) and captures precise, rule-bound maneuvers (e.g., lane changes, turns). In contrast, our work targets vehicle GPS trajectories from ITS (e.g., taxi, logistics), which is low-frequency (seconds level), sparse, and reflects long-term (minutes/hours) human mobility patterns. Moreover, our work aim to encode trajectories into semantically-rich embedding vectors, which can then be adapted to multiple downstream tasks. This pre-training paradigm represents a different technical direction from task-specific prediction models.
>
> 2. Trajectory Mamba optimizes SSM for recursive forecasting efficiency in dynamic scenes. Our Traj-Mamba Encoder, however, targets capturing long-term movement patterns of trajectories by jointly modeling and fusing GPS and road perspectives with linear time complexity: (1) Unlike directly input-dependent parameterization, it uses high-order movement behavior features to parameterize GPS-SSM, enabling precise control over how trajectory movement behavior changes affect the input selection. (2) Instead of parallel dual-view modeling followed by independent fusion module, it adopts bidirectional interaction between GPS and road views. Road-SSM parameters are extended to trajectory geographical space (rather than road network space), selecting input embeddings based on changes in trajectory movement details, while GPS intermediate embeddings are also selected via a dot-product gate at road level.
>
> **[W3]** To address the potential confounding effect of preprocessing on the soft masking mechanism, we conducted experiments without rule-based preprocessing. The results show that removing preprocessing yields comparable performance to the full model across all downstream tasks, directly validating the efficacy of the proposed soft masking mechanism.
>
>
> | Task              | DP     |       |        |        |        | ATE   |       |       | STS       |        |        |
> | ----------------- | ------ | ----- | ------ | ------ | ------ | ----- | ----- | ----- | --------- | ------ | ------ |
> | Metric            | RMSE   | MAE   | Acc@1  | Acc@5  | Recall | RMSE  | MAE   | MAPE  | Mean Rank | ACC@1  | ACC@5  |
> | w/o preprocessing | 125.56 | 86.49 | 58.31% | 86.75% | 30.52% | 50.18 | 17.41 | 4.56% | 1.22      | 96.90% | 99.00% |
> | TrajMamba         | 129.47 | 85.95 | 58.21% | 86.81% | 30.45% | 50.17 | 17.73 | 4.77% | 1.15      | 96.53% | 99.13% |
>
> Regarding the case study, due to the fact that this rebuttal can only provide textual information and is unable to offer any additional content in any form, we kindly ask for your understanding regarding our inability to present it to you here. We promise to include this experiment in subsequent versions.
>
> **[W4]** We have reproduced UniTR[3] and included it in our experiments. The results show that TrajMamba outperforms UniTR across all downstream tasks with high encoding efficiency, supporting our SOTA claim.
>
>
> | Task      | DP      |         |        |        |        | ATE    |       |        | STS       |        |        | Embed Time |
> | --------- | ------- | ------- | ------ | ------ | ------ | ------ | ----- | ------ | --------- | ------ | ------ | ---------- |
> | Metric    | RMSE    | MAE     | Acc@1  | Acc@5  | Recall | RMSE   | MAE   | MAPE   | Mean Rank | ACC@1  | ACC@5  | seconds    |
> | UniTR     | 3081.90 | 2689.76 | 27.75% | 52.41% | 6.83%  | 114.13 | 53.69 | 11.92% | 1.25      | 95.20% | 98.40% | 6.170      |
> | TrajMamba | 129.47  | 85.95   | 58.21% | 86.81% | 30.45% | 50.17  | 17.73 | 4.77%  | 1.15      | 96.53% | 99.13% | 1.721      |
>
>
> Regarding the other two works: (1) MRGRP[2] is dedicated to courier route prediction in food delivery services. It takes a set of discrete pick-up and delivery tasks as inputs and focuses on optimizing task scheduling with strict order constraints (e.g., delivery must follow pick-up for the same order). This differs fundamentally from our scenario: we process vehicle GPS trajectories and aim to learn semantically-rich embeddings for diverse downstream tasks, rather than task scheduling. (2) Path-LLM[4] studies path representation learning, where "path" refers to a theoretically navigable route on the road network, not the actual GPS trajectory generated by vehicle movement (which records real spatio-temporal behaviors). Its focus is on aligning path semantics with language models for navigation-related tasks, which does not overlap with our multi-downstream scenario. Due to these fundamental differences in input data, core objectives, and application scenarios, MRGRP and Path-LLM are not suitable as baselines for our work.
>
> **[W5]** Below we provide a detailed complexity analysis of TrajMamba's downstream components (Mask Generator + Traj-Mamba Encoder), compared with typical baselines JGRM and START.
>
> **Inference Time Complexity of TrajMamba**:
>
> 1. Symbols Definition: $d$ is the number of trajectory point features for the input mask generator, $E$ is the embedding dimension, $L$ is the number of Traj-Mamba blocks, and $n, n'$ are the lengths of the input trajectory and the compressed trajectory after processing by the Mask Generator, respectively.
> 2. In the mask generator, the time complexity is $O(nd^2)$.
> 3. For the Traj-Mamba Encoder, the time complexity is $O(n'dE)$ and $O(Ln'E^2)$ for embedding initial latent vector and $L$ stacked Traj-Mamba blocks, respectively.
> 4. The combined time complexity of TrajMamba is $O(nd^2+n'd E+Ln'E^2)$
>
> **Complexity of Typical Baselines**:
>
> - JGRM: $O(mE^2+L_rm^2E+L_f(2m)^2E)$, where $m$ is the number of unique road segments in trajectories, and $L_r, L_f$ are the numbers of route encoder layers and model interaction layers, respectively.
> - START: $O(L_gH|\mathcal{V}|E^2+L_en^2E)$, where $|\mathcal{V}|$ is the number of road segments in the road network, $H$ is the number of attention heads used by TPE-GAT, and $L_g, L_e$ are the numbers of TPE-GAT layers and TAT-Enc layers, respectively.
>
> **Conclusion**: TrajMamba achieves linear inference time complexity with respect to trajectory length $n$ by avoiding the quadratic terms present in baselines: JGRM introduces $O(m^2)$ from Transformer-based modules, while START relies on $O(n^2)$. This design ensures encoding efficiency for long or large-batch trajectories in real-world scenarios, which is crucial for real-time applications.
>
> **[Q1]** In Fig.4(b), the performance at E=256 shows significant improvement compared to E=128, but the improvement at E=512 is limited. Considering the increase in model size and encoding time resulting from changing E from 256 to 512 (which affects downstream inference efficiency and contradicts our goal of efficient inference), we chose E=256 to balance performance and efficiency.
>
> **[Q2]** We concur F1-score is a widely recognized metric for multi-class DP tasks. Our DP task involves predicting trajectory destinations at both road segment and GPS levels. For the latter, RMSE and MAE are necessary to measure prediction errors. Due to table width constraints, we selected metrics for road segment prediction following [5], and reasoned that these metrics provide complementary perspectives. To address your question, we provide supplementary F1-score results as follows, which are consistent with our Accuracy/Recall performance:
>
>
> | Dataset   | Chengdu / Xian  |
> | --------- | --------------- |
> | START     | 20.63% / 15.13% |
> | MMTEC     | 20.26% / 15.87% |
> | JGRM      | 21.35% / 16.18% |
> | TrajMamba | 27.04% / 18.50% |
>
> [5] "TrajCogn: Leveraging LLMs for Cognizing Movement Patterns and Travel Purposes from Trajectories." Proceedings of the International Joint Conference on Artificial Intelligence (IJCAI). 2025
>
> Thank you again for your earnest efforts in identifying weaknesses and raising relevant questions. We hope these supplements meet your expectations and look forward to further suggestions for improvement.

---

> > ### Comment · Reviewer_e8aq · 2025-08-05
> >
> > The authors have addressed my comments in general. I shall update my score accordingly.

---

> > > ### Author Response · Authors · 2025-08-06
> > >
> > > We're glad that our response has gained your recognition, and truly appreciate your time and feedback. Please feel free to reach out with any further concerns, and we look forward to your continued suggestions for improvement.

---

### Official Review · Reviewer_7HKs · 2025-07-02

**Clarity:** 3
**Significance:** 4
**Originality:** 4
**Rating:** 6
**Confidence:** 5

**Summary:**

The paper introduces TrajMamba, a Mamba-based framework for trajectory compression and embedding. The authors propose a Travel Purpose-aware Pre-training method to incorporate travel purpose information, thereby alleviating the burden of encoding textual descriptions through contrastive learning. Additionally, a Knowledge Distillation Pre-training scheme is introduced to explicitly eliminate redundant trajectory points using a selfsupervised paradigm. Experiments conducted on two real-world datasets demonstrate the effectiveness of the proposed model.

**Questions:**

See weaknesses.

**Ethical Concerns:**

["NO or VERY MINOR ethics concerns only"]

**Final Justification:**

The authors has addressed most of my previous concerns. Combing with other reviewers' comments, I decide to raise my score to 6.

**Limitations:**

Yes.

**Quality:**

3

**Strengths And Weaknesses:**

Strengths:
[S1] Using sentence embedding models and contrastive learning to capture trajectory travel intentions is both novel and promising. By shifting the computational burden to the offline phase, the training efficiency is greatly improved.
[S2] The approach of using a mask and the Gaussian error function for self-supervised trajectory compression training is ingenious.
[S3] The experimental setup is reasonable, and both the effectiveness and efficiency of the model are validated.

Weaknesses:
[W1] The discussion on trajectory compression is insufficient. Is there any analysis of the compressed trajectories, such as the average compression ratio? How were the weights (1/2) for the loss functions L^MEC and L^mask determined, and how do different weights affect downstream tasks and compression performance?
[W2] There is a lack of discussion regarding the sentence embedding. The paper uses text-embedding-3-large; would using other models affect the performance of the proposed approach？

---

> ### Author Rebuttal · Authors · 2025-07-31
>
> We sincerely appreciate your recognition of our work—this means a great deal to us. Below, we address your doubts as follows.
>
> **[W1]** We provide the following supplementary analysis:
>
> - The average compression ratio of trajectories (=compressed length/raw length) on Chengdu and Xian is shown below. This result proves that our model successfully identifies and removes a considerable part of the implicit redundancy contained in trajectories, retaining key points that benefit trajectory embedding representation.
>
>   | Dataset | ratio after preprocessing | ratio after soft masking mechanism |
>   | ------- | -------------------------------------------------- | ------------------------------------------------------------ |
>   | Chengdu | 67.189%                                            | 42.159%                                                      |
>   | Xian    | 70.419%                                            | 35.094%                                                      |
>
> - We set the weights for the loss functions $L^{MEC}$ and $L^{mask}$ to 1/2 to balance these two objectives. A larger weight for $L^{MEC}$ results in compressed representations containing more information from the full-trajectory embedding; a larger weight for $L^{mask}$ produces shorter compressed trajectories from the mask generator, achieving greater compression. We conducted supplementary experiments on the Chengdu dataset to explore the impact of different weights on downstream tasks and compression performance. The results demonstrate that when the weights for ($L^{MEC}$, $L^{mask}$) are **(0.5, 0.5)**, our model achieves the best overall performance across all downstream tasks and encoding time.
>
>   | Task                  |             | Destination | Prediction  |             |             | Arrival    | Time       | Estimation | Similar   | Trajectory  | Search      | Embed Time |
>   | --------------------- | ----------- | ----------- | ----------- | ----------- | ----------- | ---------- | ---------- | ---------- | --------- | ----------- | ----------- | ---------- |
>   | Metric                | RMSE        | MAE         | Acc@1       | Acc@5       | Recall      | RMSE       | MAE        | MAPE       | Mean Rank | ACC@1       | ACC@5       | seconds    |
>   | (1.0, 0.0)            | **125.619** | 86.543      | 56.277%     | 83.854%     | 26.363%     | 51.766     | 19.657     | 5.335%     | 1.662     | 95.800%     | 98.300%     | 1.944      |
>   | (0.7, 0.3)            | 135.447     | 91.966      | 58.130%     | 86.362%     | 30.075%     | 54.509     | 17.940     | 4.802%     | 1.155     | **96.600%** | 98.900%     | 1.731      |
>   | (0.3, 0.7)            | 134.870     | 90.072      | 58.145%     | 86.546%     | 30.031%     | 50.978     | 18.556     | 5.053%     | 1.204     | 95.800%     | 98.800%     | **1.692**  |
>   | (**0.5, 0.5**) (Ours) | 129.474     | **85.947**  | **58.205%** | **86.813%** | **30.446%** | **50.166** | **17.727** | **4.772%** | **1.148** | 96.525%     | **99.125%** | 1.721      |
>
> **[W2]** We evaluated two additional pre-trained textual embedding modules from HuggingFace: `hfl/chinese-lert-large` and `BAAI/bge-large-zh-noinstruct`. We explored how different textual embedding modules affect our model's performance. The results show that using alternative models has limited impact on performance, demonstrating that our pre-training process enables the model to learn robust representations.
>
> | Task                    |             | Destination | Prediction  |             |             | Arrival    | Time       | Estimation | Similar   | Trajectory  | Search      |
> | ----------------------- | ----------- | ----------- | ----------- | ----------- | ----------- | ---------- | ---------- | ---------- | --------- | ----------- | ----------- |
> | Metric                  | RMSE        | MAE         | Acc@1       | Acc@5       | Recall      | RMSE       | MAE        | MAPE       | Mean Rank | ACC@1       | ACC@5       |
> | chinese-lert-large      | **127.967** | **83.727**  | 58.222%     | 86.717%     | **30.579%** | 50.245     | 17.755     | 4.653%     | **1.131** | **97.300%** | 99.300%     |
> | bge-large-zh-noinstruct | 131.338     | 87.626      | **58.390%** | 86.799%     | 30.298%     | 50.317     | **17.458** | **4.631%** | 1.153     | 96.300%     | **99.400%** |
> | TrajMamba               | 129.474     | 85.947      | 58.205%     | **86.813%** | 30.446%     | **50.166** | 17.727     | 4.772%     | 1.148     | 96.525%     | 99.125%     |
>
>
> We sincerely hope that the above content has adequately addressed your questions and concerns. Your positive feedback has been highly encouraging, and we look forward to further enhancing the work with your continued guidance.

---

> > ### Comment · Reviewer_7HKs · 2025-08-03
> > **sufficient rebuttal**
> >
> > The authors has addressed most of my previous concerns. By considering with other reviewers' comments, I decide to raise my score to 6.

---

> > > ### Author Response · Authors · 2025-08-04
> > >
> > > Thank you again for your recognition of our work, and if you have any other questions, please feel free to contact us.

---

### Decision · Program_Chairs · 2025-09-17

**Decision:**

Accept (spotlight)

**Comment:**

The paper introduces TrajMamba, a Mamba-based framework for trajectory compression and embedding. It has a Travel Purpose-aware Pre-training method to incorporate travel purpose information. This approach alleviates the burden of encoding textual descriptions through contrastive learning. In addition, it introduces a Knowledge Distillation Pre-training scheme to remove redundant trajectory points via self-supervised learning. Experiments demonstrate the effectiveness of the proposed approach.

All reviewers agreed to accept the paper after rebuttal. The AC agrees with the decision and recommends it as  a highlight, given its potential to guide future research on large-scale trajectory representation learning.